



# A global model perturbed parameter ensemble study of secondary organic aerosol formation

Kamalika Sengupta[1], Kirsty Pringle[1], Jill S. Johnson[1], Carly Reddington[1], Jo Browse[2], Catherine E. Scott[1], and Ken Carslaw[1]

[1]Institute for Climate and Atmospheric Science, School of Earth and Environment, University of Leeds, Leeds, UK
[2]Center for Geography and Environmental Science, University of Exeter, Penryn, Cornwall, UK

*Correspondence to:* Kamalika Sengupta (k.sengupta@leeds.ac.uk)

**Abstract.** A global model perturbed parameter ensemble of 60 simulations was used to explore how combinations of six parameters related to secondary organic aerosol (SOA) formation affect particle number concentrations and organic aerosol mass. The parameters represent the formation of organic compounds with different volatilities from biogenic and anthropogenic sources. The most plausible parameter combinations were deter-

5 mined by comparing the simulations against observations of the number concentration of particles larger than 3 nm diameter (N3), the number concentration of particles larger than 50 nm diameter (N50), and the organic aerosol (OA) mass concentration. The simulations expose a high degree of model equifinality in which the skill of widely different parameter combinations cannot be distinguished against observations. We therefore conclude that, based on the observations we have used, a 6-parameter SOA scheme is under-determined. Nevertheless, the

10 model skill in simulating N3 and N50 is clearly determined by the low and extremely low volatility compounds that affect new particle formation and growth, and the skill in simulating OA mass is determined by the low volatility and semi-volatile compounds. The biogenic low volatility class of compounds that grow nucleated clusters and condense on all particles is found to have the strongest effect on the model skill in simulating N3, N50 and OA. The simulations also expose potential structural deficiencies in the model: we find that parameter

combinations that are best for N3 and N50 are worst for OA mass, and the ensemble exaggerates the observed seasonal cycle of particle concentrations − a deficiency that we conclude requires an additional anthropogenic source of either primary or secondary particles.





*Copyright statement.* TEXT

## 1   Introduction

About 20-50% of lower tropospheric fine aerosol mass in continental mid latitudes and almost 90% in the tropics is composed of organic material (Kanakidou et al., 2005). The major fraction of organic aerosol has been found to be secondary (Zhang et al., 2007; Jimenez et al., 2009), formed as a result of atmospheric oxidation of volatile organic compounds (VOCs) leading to secondary organic aerosol (SOA). Estimating atmospheric SOA is important for accurately estimating the anthropogenic aerosol radiative forcing (Maria et al., 2004; Tsigaridis and Kanakidou, 2007; Carslaw et al., 2010; Riipinen et al., 2011; Makkonen et al., 2012; Shrivastava et al., 2017; Tsigaridis and Kanakidou, 2018). Despite the importance of SOA for climate, a comprehensive characterisation of atmospheric VOCs, their reaction pathways, and their SOA formation potential has not yet been possible. Characterisation of VOCs is challenging because of the very large number of compounds involved and their diverse sources; tens of thousands of VOCs have been identified in the atmosphere and yet more still remain to be detected (Goldstein and Galbally, 2007). For the SOA precursor gases that have been identified, questions remain about their emission sources, their chemical conversion to SOA, and the effects of atmospheric chemical composition and oxidants on SOA formation.

From a modelling perspective a further challenge is how to deal with the enormous chemical complexity of the system to adequately parameterise laboratory and observational results and incorporate them in large-scale models. Models traditionally use laboratory measurements of reaction rate constants and product yields to calculate the production of highly oxidised VOCs from the reaction between precursor VOCs and atmospheric oxidants. Only a very small range of natural VOCs are accounted for, such as monoterpenes or isoprene. Uptake of SOA on to particles is then simulated either time-dependently (often called kinetic uptake) or assuming thermodynamic partitioning (Riipinen et al., 2012). Such 'bottom-up' approaches to simulating SOA predict a global SOA budget at the lower end of the total uncertainty range (Kanakidou et al., 2005), most likely because simulations do not capture the full range of VOCs and atmospheric oxidative pathways that lead to a range of products of varying volatilities (Hallquist et al., 2009; Tsigaridis et al., 2014). Studies like Heald et al. (2005);





*1 INTRODUCTION*

Johnson et al. (2006); Spracklen et al. (2011) show that models have consistently and significantly underesti-mated SOA concentrations in different parts of the atmosphere. Tsigaridis et al. (2014) shows models largely underestimate the amount of organic aerosol present in the atmosphere with the underestimation being strongest in urban regions based on a study involving 31 global models.

It is well-established that atmospheric organic molecules strongly affect the number concentrations of climate-relevant sized particles by condensing on and growing aerosol particles (Riipinen et al., 2011) or by promoting particle formation (O'Dowd et al., 2002; Zhang et al., 2004; Metzger et al., 2010). Recent studies (Ehn et al., 2014; Jokinen et al., 2015; Kirkby et al., 2016; Tröstl et al., 2016; Bianchi et al., 2019) have established the

importance of atmospheric highly oxygenated organic molecules in the formation and growth of new aerosol particles. Subsequent model simulations based on these experimental data have shown that new particle forma-tion involving organic molecules could explain the seasonal cycle of particle concentrations (Riccobono et al., 2014) and provide a source of aerosol in clean pre-industrial environments that is important for climate (Gordon et al., 2016). It has been estimated that global cloud condensation nuclei (at 0.2% supersaturation) would be

about one-quarter lower without biogenic VOCs and about three-quarters of this effect is caused by the role of organic molecules in nucleation and early growth (Gordon et al., 2017).

The importance of SOA for climate means that large-scale models need to simulate the contributions of these highly oxygenated molecules in nucleation or subsequent particle growth. Several modelling studies (such as

Farina et al., 2010; Pye and Seinfeld, 2010; Jathar et al., 2011) have implemented the Volatility Basis Set (VBS) framework proposed by Donahue et al. (2012) for the description of organic partitioning and chemical ageing. Other SOA modelling frameworks have been proposed by Odum et al. (1996); Camredon et al. (2007); Parikh et al. (2011). However increased model complexity inevitably requires more computational resources and more runtime, both of which need to be considered carefully by modellers. Shrivastava et al. (2011) found comparable

predictions of observed OA between a 9-species VBS approach and a 2-species simplified VBS approach, the latter being a factor 2 lower in computational cost. Shrivastava et al. (2011) concluded the 2-species approach is well-suited to represent the complex evolution of atmospheric organic aerosols. Riipinen et al. (2011) and Scott et al. (2015) propose simplified SOA formation scheme representing species of two volatilities. Tsigaridis et al.





(2014) shows global model skill does not increase with model complexity with regard to organic aerosol mass concentrations.

There is the risk that added complexities in models that are not well-constrained by experimental data could
increase model uncertainty and thereby introduce more uncertainty in the quantification of the effect of anthropogenic aerosols (Lee et al., 2016). Although model complexity is increased in order to improve the representation of the physical, chemical or optical properties of SOA (Tsigaridis et al., 2014), a more-complex model that matches some observations may not have lower uncertainty in making predictions because of the increased likelihood of compensating errors, often called model equifinality (Beven, 2006). The simulated particle number
(and anthropogenic aerosol forcing) is affected by several model parameters and their associated uncertainties, some of which may be compensating for each other. Tuning any one parameter within the model (e.g. the nucleation rate) to improve the model performance against observations of one aspect of the atmospheric aerosol distribution (e.g. the total particle number concentration), may adversely affect the model performance in other outputs (e.g. total aerosol mass). Such tuned observationally constrained models give the impression of low
aerosol uncertainty and model robustness but still predict a large range of aerosol forcing in Lee et al. (2016).

In this paper we use a perturbed parameter ensemble of 60 simulations of a global aerosol model to examine the effect of six uncertain parameters in SOA formation on simulated organic aerosol mass and particle number concentrations. We compare three model outputs against observations: the number concentration of
particles larger than 3 nm dry diameter (N3) and 50 nm dry diameter (N50), and the mass concentration of organic aerosol (OA). Our primary aim is to understand the sensitivity of N3, N50 and OA to the combinations of model input parameters that control the formation of SOA. We identify parts of parameter space that result in the best agreement with observations of N3, N50 and OA. Further, we identify parameter combinations that produce models that are indistinguishable in terms of their simulations of number and mass concentrations.





## 2   Methods

### 2.1   GLOMAP global aerosol model

We use the GLOMAP (GLObal Model of Aerosol Processes) modal aerosol microphysics model (Mann et al., 2010), which is an extension to the TOMCAT 3-D chemical transport model (Chipperfield, 2006). The model has a horizontal resolution of 2.8° x 2.8°, with 31 hybrid $\sigma$-pressure levels from surface to 10 hPa. Large-scale atmospheric transport in the model for 2008 is driven by ERA-Interim reanalysis produced by the European Centre for Medium-Range Weather Forecasts (ECMWF) at 6 hourly intervals. The aerosol distribution is simulated using four hydrophilic modes (nucleation, Aitken, accumulation and coarse) and one non-hydrophilic Aitken mode. The aerosol phase has four components: sulphate, sea salt, black carbon and organic carbon. Where more than one species is contained in a mode we assume internal mixing (Mann et al., 2010).

### 2.2   SOA scheme

The model described in Mann et al. (2010) includes one SOA species produced from oxidation of monoterpenes only. In this study we produce six SOA species from oxygenated organic compounds derived from the oxidation of monoterpene, isoprene and anthropogenic sources. Monthly monoterpene and isoprene emissions used in the model are generated from the Community Land Model by Sarah Monks (MEGANv2.1; Guenther et al., 2012) Emission of anthropogenic VOC is parameterised using the approach in Spracklen et al. (2011) − we use CO emissions from the MACCity inventory and assume a SOA/OC mass ratio of 1.4. Atmospheric oxidants (OH$^{.}$, O$_3$, NO$_3$) are taken from 6-hourly monthly mean values calculated offline from a TOMCAT simulation and interpolated to the model chemical time step (Monks et al., 2017).

Figure 1 shows a schematic of the treatment of SOA in this study. Gas-phase oxygenated organic compounds (ox-VOC) are represented by three classes: Extremely Low Volatility (ELVOC), Low Volatility (LVOC) and Semi-volatile (SVOC) organic compounds.





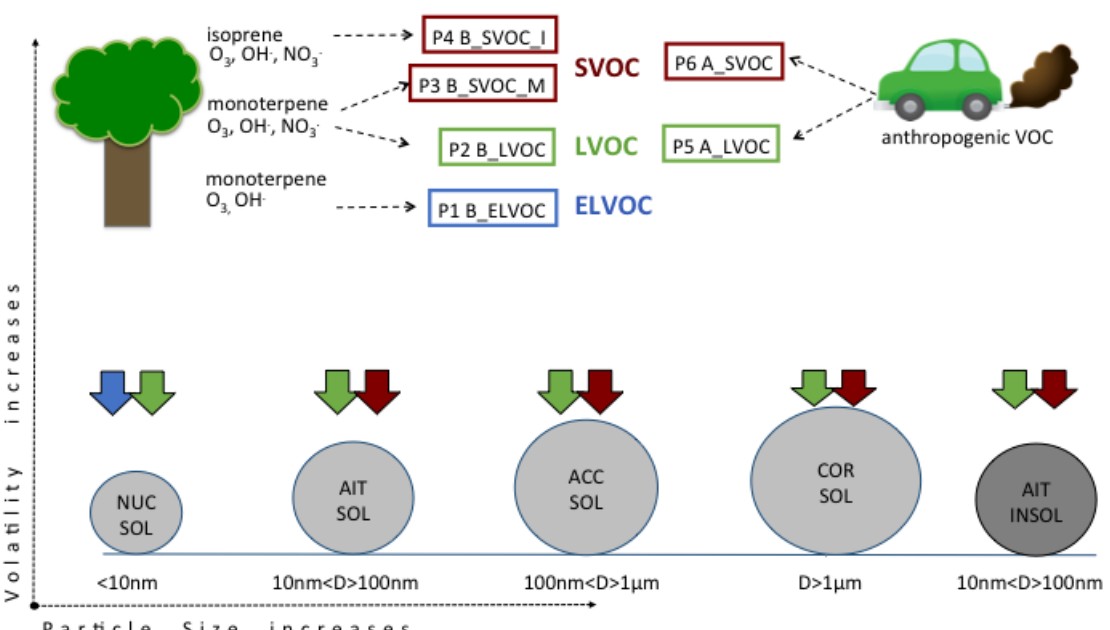

**Figure 1.** Schematic showing the SOA formation scheme in GLOMAP-mode and the six oxidised VOCs (products of photochemical oxidation of emitted VOCs that eventually produce SOA) whose concentrations are perturbed in this study. The six oxidised VOCs represent three volatility categories - extremely low volatility organic compounds (ELVOC, blue), low volatility organic compounds (LVOC, green) and semi-volatile organic compounds (SVOC, red). Prefix 'A' indicates precursor gases of anthropogenic origin and 'B' indicates biogenic origin. The schematic shows the precursor gases and oxidants that react to produce these ox-VOCs, their relative volatility (ELVOC<LVOC<SVOC) and the mechanism (nucleation for ELVOC, kinetic condensation for LVOCs and mass-based partitioning for SVOCs) by which they add to the condensed phase (represented here by the five modes:- nucleation soluble, Aitken soluble, accumulation soluble, coarse soluble and Aitken insoluble modes).

The ELVOCs are assumed to derive only from biogenic sources and nucleate to form new particles that either

5   grow or are lost to pre-existing larger particles (Gordon et al., 2016). The LVOCs are assumed to condense kinetically and irreversibly (i.e., with zero vapour pressure) on all particles, and the SVOCs are assumed to partition into all particles, except those in the nucleation mode, in proportion to the pre-existing organic mass in the mode (Scott et al., 2015). The LVOC and SVOC are further divided into biogenic (prefix B) and anthropogenic (prefix



A). The biogenic precursors are split into monoterpenes (suffix M) and isoprene (suffix I), with the monoter-

10   penes allowed to form both LVOC and SVOC and the isoprene forming only SVOC. The precursors, oxidants

and role of each ox-VOC in SOA formation are defined in Table 1. The ox-VOCs in this new scheme of SOA

formation produced from bimolecular oxidation reactions of VOCs are, B_ELVOC, B_LVOC, B_SVOC_M,

B_SVOC_I, A_LVOC and A_SVOC. The six perturbed parameters in this study are scaling factors or reaction

yields that control the concentrations of these six ox-VOCs.

| ox-VOC | Volatility | Nature of source | Role | Parent VOC | Oxidants |
|---|---|---|---|---|---|
| B_ELVOC | Extremely Low | biogenic | nucleation | $\alpha$-pinene | $O_3$, $OH^{\cdot}$ |
| B_LVOC | Low | biogenic | kinetic condensation | $\alpha$-pinene | $O_3$, $OH^{\cdot}$, $NO_3$ |
| B_SVOC_M | Semi | biogenic | mass based partition | $\alpha$-pinene | $O_3$, $OH^{\cdot}$, $NO_3$ |
| B_SVOC_I | Semi | biogenic | mass based partition | isoprene | $O_3$, $OH^{\cdot}$, $NO_3$ |
| A_LVOC | Low | anthropogenic | kinetic condensation | anthropogenic VOCs | $OH^{\cdot}$ |
| A_SVOC | Semi | anthropogenic | mass based partition | anthropogenic VOCs | $OH^{\cdot}$ |

**Table 1.** List of oxidised VOCs implemented in this study, the volatility class they represent, whether produced from biogenic or anthropogenic sources, how they take part in atmospheric SOA formation, parent VOC, and oxidants that react to produce each ox-VOC. MT stands for monoterpene, IP stands for isoprene and CO stands for carbon monoxide indicating anthropogenically sourced VOC.

## 2.3   Perturbed parameter ensemble

Simultaneous perturbation of six model parameters within each of their chosen range forms a 6-D parameter

5   space within which we explored the competing and compensating effects of these parameters on model outputs.

An ensemble of 60 simulations was produced, each with a different combination of the six parameters. Simula-

tions were run for the year 2008. To ensure good coverage of the 6-D parameter uncertainty space, the maximin

Latin hypercube sampling method (McKay et al., 1979) was used to choose the combinations of parameters





(Lee et al., 2011, 2013). The relative variation of the ox-VOCs in each simulation is shown in Figure 2.

We perturb the concentrations of SOA-producing ox-VOCs by perturbing the yields of bimolecular oxidation reactions for B_LVOC, B_SVOC_M, B_SVOC_I, A_LVOC and A_SVOC. The baseline molar yields for each of these ox-VOCs before perturbation were: 13% for B_LVOC, 13% for B_SVOC_M, 3% for B_SVOC_I producing approximately 40 Tg yr$^{-1}$ of SOA from biogenic sources. The total anthropogenic ox-VOCs are split

15 equally between A_LVOC and A_SVOC (effectively a 50% yield of the total anthropogenic each) together producing approximately 63 Tg yr$^{-1}$ of SOA. Within the ensemble the yield of each of the above is perturbed from 0 to 20 times the baseline for biogenic ox-VOCs (B_LVOC, B_SVOC_M, B_SVOC_I) and from 0 to 5 times the baseline for anthropogenic ox-VOCs (A_LVOC and A_SVOC). We perturb the concentrations of ox-VOCs taking part in the model nucleation scheme, B_ELVOC, using a scaling factor between 0 and 10, where 0 is

20 equivalent to sulphuric acid-only nucleation and 10 being equivalent to ELVOC concentrations about a factor of 10 higher than reported in Kirkby et al. (2016). The ranges (Table 2) were chosen to encompass a wide range of uncertainties and simplifications in the model. These include VOC emission uncertainties; structural uncertainty such as neglected precursor gases (e.g., sesquiterpenes); uncertainty in yields of the ox-VOCs from oxidation reactions; simplifications to the oxidation pathways (in GLOMAP-mode only single-stage oxidation products are represented); and uncertainty in SOA due to neglecting the volatility distribution and re-evaporation of SOA (Donahue et al., 2011, 2012). Changing the ox-VOC concentrations by changing the yields of chemical reactions has the same effect in the model as perturbing the emissions of the parent VOCs. A yield above 100% in

the bimolecular reactions should therefore be interpreted as an increase in the total concentration of reactants. The advantage of varying the yield of ox-VOCs rather than the emissions of VOCs is that a perturbation applied to one ox-VOC does not affect the production or loss of the other ox-VOCs when they have the same parent VOC (i.e., they have uncorrelated effects across the 6-D parameter space).





| Perturbed Parameter | | | Perturbation Range | | | Default SOA produced |
|---|---|---|---|---|---|---|
| ox-VOC | Produced from | Default yield value (in%) | minimum | maximum | Perturbation | Tg(SOA) yr$^{-1}$ |
| B_ELVOC | Monoterpene, $O_3$ | 3.2 | 0 | 10×default | scaled | - |
|  | Monoterpene, OH· | 1.2 | 0 | 10×default | scaled | - |
| B_LVOC | Monoterpene, $O_3$ | 13 | 0 | 20×default | absolute | |
|  | Monoterpene, OH· | 13 | 0 | 20×default | absolute | 16 |
|  | Monoterpene, $NO_3$ | 13 | 0 | 20×default | absolute | |
| B_SVOC_M | Monoterpene, $O_3$ | 13 | 0 | 20×default | absolute | |
|  | Monoterpene, OH· | 13 | 0 | 20×default | absolute | 17 |
|  | Monoterpene, $NO_3$ | 13 | 0 | 20×default | absolute | |
| B_SVOC_I | Isoprene, $O_3$ | 3 | 0 | 20×default | absolute | |
|  | Isoprene, OH· | 3 | 0 | 20×default | absolute | 6 |
|  | Isoprene, $NO_3$ | 3 | 0 | 20×default | absolute | |
| A_LVOC | Anthropogenic VOC, OH· | 50 | 0 | 5×default | absolute | 34 |
| A_SVOC | Anthropogenic VOC, OH· | 50 | 0 | 5×default | absolute | 34 |

**Table 2.** Perturbed parameters and ranges. Also listed are the precursor VOC gases that produce the ox-VOCs, the value of ox-VOC yields in the default model version (unperturbed), how the perturbation for each parameter is implemented in the model ('absolute' for replacing the default yield value by the perturbation, 'scaled' for scaling the default yield value by the perturbation), and the amount of SOA produced from each parameter in the default model version.







**Figure 2.** The relative variation of the six perturbed parameters (in%) for each ensemble member (numbered 1 to 60). Each hexagon (grey dashed area) represents the 6-D parameter space and the positions of the black dots show the position of each parameter within its range for the specific ensemble member. The dots are joined and shaded green for easy identification of explored parameter space in each ensemble member. Counter clockwise from top, the black dots represent parameter settings for B_ELVOC, B_LVOC, B_SVOC_M, B_SVOC_I, A_LVOC and A_SVOC respectively. Example interpretation: in simulation 19 (fourth row, 1st hexagon) B_SVOC_I and A_SVOC concentrations are towards the lower ends of the respective ranges being explored for each of them while concentrations of A_LVOC is towards the high end of the A_LVOC range.





## 2.4 Observations

Figure 3 shows a map of ground-based observation stations used to compare the surface-level number concentrations of particles with dry diameter greater than 3 nm (N3 in $cm^{-3}$), particles with dry diameter greater than 50 nm (N50 in $cm^{-3}$) and the mass concentration of organic aerosol (OA in $\mu g\, m^{-3}$) predicted by the ensemble.

N3 observations cover 34 ground stations worldwide as used in Spracklen et al. (2010). Measurements of N3
were made between 1994 and 2009 using either condensation particle counters (CPCs), scanning mobility particle sizers (SMPS), differential mobility particle sizers (DMPS) or Diffusion Aerosol Spectroscopes (DAS). The N3 data set is fully described in Spracklen et al. (2010) and has been used in previous studies such as Riccobono et al. (2014), Gordon et al. (2016) and Dunne et al. (2016). N50 observations are from 31 ground stations worldwide including sites in Europe as described in Asmi et al. (2011) and African(Vakkari et al., 2013),
Indian(Hyvärinen et al., 2010), Canadian(Jeong et al., 2010; Takahama et al., 2011) and polar(Asmi et al., 2016; Hansen et al., 2009) sites obtained from individual projects and online data portals. Measurements of N50 were made between 2007 and 2015 using DMPS or SMPS instruments. OA observations cover 41 ground stations worldwide from the Global Aerosol Synthesis and Science Project (GASSP) database (Reddington et al., 2017). OA measurements were made between 1990 and 2015 using the Aerosol Mass Spectrometer (AMS) and the associated Aerosol Chemical Speciation Monitor (ACSM) that characterise the mass and chemical composition of particulate matter (Canagaratna et al., 2007; Ng et al., 2011).

All station data were averaged to create monthly mean values. Station heights were matched to model pressure levels for each month using barometric altitude. Stations cover a wide range of atmospheric conditions
5 such as continental boundary layer (CBL e.g. Hyytiala, Harwell, Botsalano), marine boundary layer (MBL e.g. Mace Head, Trinidad Head, Sable Island) and free tropospheric (FT e.g. Nepal, Jungfraujoch, Pico Espejo, Mauna Loa) sites. Errors in measurements are estimated to be around 30% on average, depending strongly on the spatial heterogeneity of sources (Reddington et al., 2017).







**Figure 3.** Locations of ground-based sites where model-observation match is compared for N3 (34 locations, symbols in light), N50 (31 locations, symbols in blue) and OA (41 locations, symbols in red).

10    To examine the performance of the ensemble members we use statistical metrics including correlation coefficient, normalised mean bias factor and Taylor Skill Score or TSS (Taylor, 2001), in the following sections.




NMBF is an unbiased and symmetric metric with a range from -∞ to +∞, with 0 corresponding to exact agreement. It is calculated using the following equation:

$$NMBF = \begin{cases} \dfrac{\sum\limits_{i=1}^{N}(M_i - O_i)}{\sum\limits_{i=1}^{N} O_i}, & if\ \bar{M} \geq \bar{O} \\[2em] \dfrac{\sum\limits_{i=1}^{N}(M_i - O_i)}{\sum\limits_{i=1}^{N} M_i}, & if\ \bar{M} < \bar{O} \end{cases} \tag{1}$$

where for ensemble members $i = \{1, 2, ...N\}$, $M_i$ and $O_i$ are the modelled and observed variables, the $\bar{M}$ represents the mean across all ensemble members and $\bar{O}$ represents mean across all observations. $NMBF = 1.5(-1.5)$ means the model is biased towards overestimating (underestimating) observations by a factor of 2.5 ($NMBF$+1).

The Taylor Skill Score (TSS) is calculated as:

$$TSS = \frac{4(1+R)^4}{(\hat{\sigma}_f + \frac{1}{\hat{\sigma}_f})^2 (1+R_0)^4} \tag{2}$$

where $\hat{\sigma}$ is the normalised standard deviation ($\sigma_{model}/\sigma_{observation}$) and $R_0$ is the maximum correlation attainable by the model, given the internal variability in the system (here $R_0$ is assumed to be 1). As the model

variance approaches the variance in the observations and R approaches $R_0$, TSS approaches unity. As the model variance approaches zero or the correlation becomes more and more negative, TSS approaches zero.





## 3   Results

### 3.1   Global and regional aerosol mass and number

The global mass of SOA produced in the model simulations ranges from 220 to 850 Tg yr$^{-1}$ across the 6-D

parameter range. Our global SOA range covers the upper end of the $50-380$ Tg yr$^{-1}$ global SOA found in
Spracklen et al. (2011) after constraint of an earlier version of the GLOMAP model to global AMS OA ob-
servations. Although individual ox-VOC yields were varied between 0 and 20 times their baseline yields, the
total global production of SOA varies by only a factor of 4 and the lowest value is 220 Tg yr$^{-1}$. Global mean
N50 varies by a factor of 2.39 within the ensemble (from 377 to 903 cm$^{-3}$). Modelled N3 is more sensitive to

changes in ox-VOCs than N50; global mean values vary by a factor of 3.5 across the sixty ensemble members
(from 531 to 1889 cm$^{-3}$).

Figure 4 shows that the ensemble members produce large regional variations in OA, even when they predict
similar global values. For example, simulations 9 (subplot 3,4) and 36 (subplot 3,5) have similar global mean

OA concentrations (6.47 and 6.49 $\mu$g m$^{-3}$, respectively) but they simulate very different OA over the highly
polluted regions of South Asia. Such regional variations are dependent on the parameter settings of the ox-
VOCs. In this case simulations 9 and 36 particularly differ in the contribution from B_SVOC_M and A_LVOC.
Modelling efforts need to focus on capturing the competing and compensating effects of ox-VOCs contributing
to different stages of particle formation and growth, rather than detailed representation of any one contributions.
This also emphasises the need to compare model outputs with regional as well as global metrics and observa-
tions to determine if the model is performing well.

Simulations 45 (subplot 9,2) and 49 (subplot 10,5) demonstrate the role of anthropogenic VOCs on the sim-
ulated aerosol size distribution and OA mass. Global mean N3, N50 and OA in simulations 45 and 49 are all in
the upper quartile of the ensemble's output distribution for these quantities. Both of these simulations have high
concentrations of B_ELVOC, which promote nucleation, and moderate to high B_LVOC, which promotes the

10   survival of nucleated particles, contributing to the high global mean particle number and mass concentrations. In
simulation 45 both biogenic and anthropogenic ox-VOCs contribute significantly with B_ELVOC, B_SVOC_I



and A_LVOC concentrations being predominant (Figure 2). In contrast contributions to SOA are predominantly biogenic in simulation 49 (Figure 2). This difference in parameter design is reflected in the global distributions. Simulation 49 with low anthropogenic contributions predicts lower OA concentrations in the highly polluted SE Asian region. Anthropogenic ox-VOCs in the model favour the loss of smaller particles (which are more susceptible to condensation sinks) and the growth of larger particles (mass-based partitioning of A_SVOC).

Therefore in the simulation with low anthropogenic SOA more smaller particles survive but fewer particles hold substantial mass leading to lower OA mass concentrations. This is also supported by the considerably higher N3 but only slightly higher N50 number concentrations in the SE Asian region in simulation 49 compared to simulation 45. Anthropogenic sources of SOA affect the pre-industrial and present-day atmospheres differently in global models (Carslaw et al., 2013). Under-represented anthropogenic sources in global models therefore

have greater climatic implications than under-represented biogenic SOA sources.



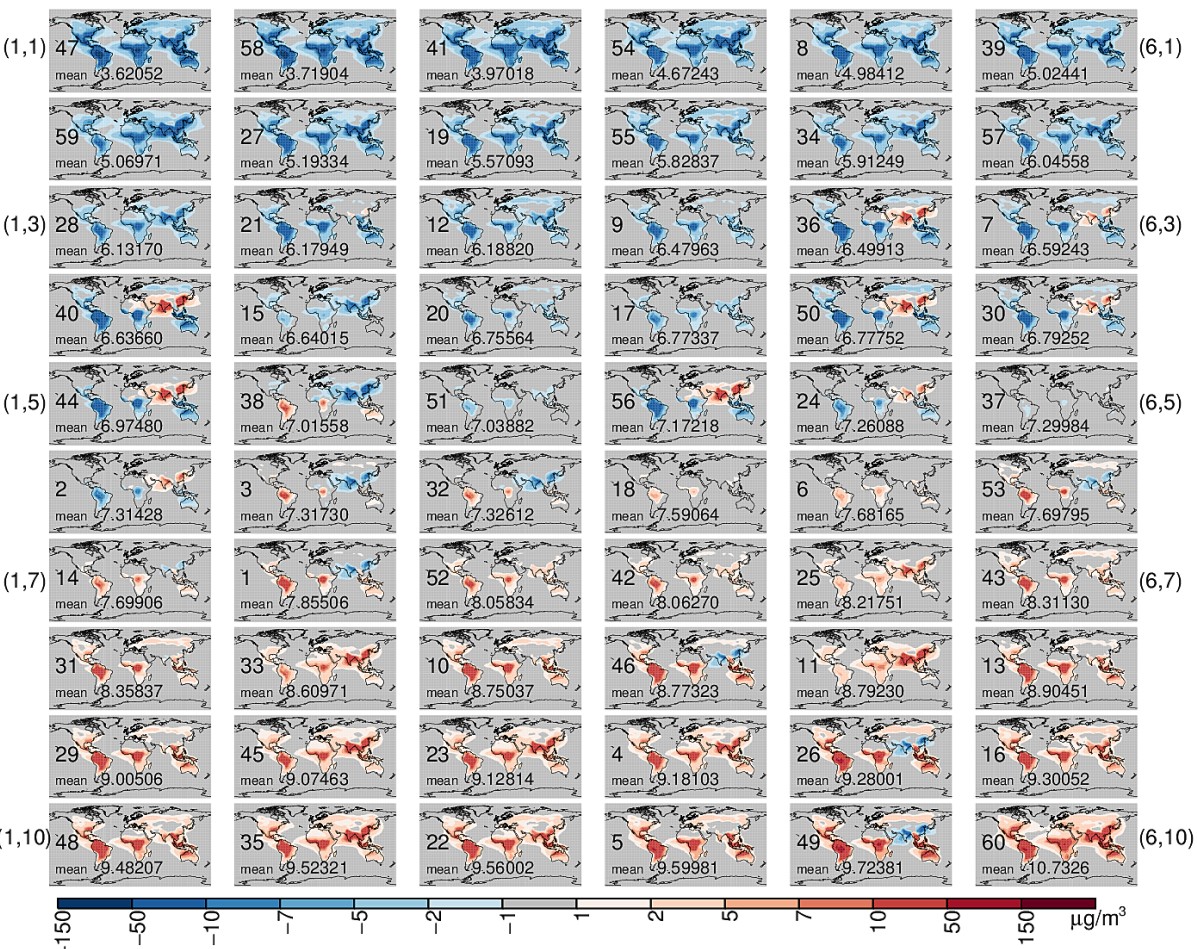

**Figure 4.** Global annual mean anomaly of organic aerosol mass at the surface (OA in $\mu$g m$^{-3}$) produced within the ensemble. Each subplot shows the anomaly of an ensemble member (numbered between 1 and 60) from the ensemble mean OA. The global mean OA is given on each subplot. The subplots are arranged in order of increasing global mean OA.





**Figure 5.** Global annual mean anomaly of N3 number concentration at the surface (cm$^{-3}$) produced within the ensemble. Each subplot shows the anomaly of an ensemble member (numbered between 1 and 60) from the ensemble mean N3. The global mean N3 is given on each subplot. The subplots are arranged in order of increasing global mean N3.



**Figure 6.** Global annual mean anomaly of N50 number concentration at the surface (cm$^{-3}$) produced within the ensemble. Each subplot shows the anomaly of an ensemble member (numbered between 1 and 60) from the ensemble mean N50. The global mean N50 is given on each subplot. The subplots are arranged in order of increasing global mean N3.





Figures 5 and 6 show the global distribution of N50 and N3. The panels in each figure are ordered according to increasing global mean N3. There is a general increase in N50 with increasing N3, but eleven of the simulations clearly show low N50 concentrations despite high N3 (simulations 14, 39, 47, 56, 50, 38, 9, 15, 15, 59, 51 in Figure 6). All of these simulations have one aspect in common − very low yields of B_LVOC (Figure 2). High concentrations of nucleating B_ELVOC in these simulations initially facilitates the formation of particles, but low yields of B_LVOC, especially when coupled with high B_SVOC_I and A_LVOC, suppresses particle growth up to 50 nm.

In contrast simulations 13, 24, 35 and 46 have low concentrations of B_ELVOC but relatively high B_LVOC. Despite the low B_ELVOC concentrations, which produces fewer nucleated particles, the relatively high concentration of B_LVOC ensures that more of the nucleated particles reach 50 nm diameter in these simulations. Consequently for these simulations the simulated global mean N3 concentrations are in the lower quartile within the ensemble but global mean N50 concentrations are in the interquartile range within the ensemble. B_LVOC can compensate for B_ELVOC to some extent and clearly stands out as the most important controller of climate-relevant particle number concentrations.

Figure 7 shows how the global mean N50 and OA concentrations depend on the six ox-VOC parameters. The shading, blue for OA and red for N50, indicate whether the global mean values within the ensemble fall in the upper quartile (Q4 indicated by dark shade), inter-quartile range (IQR indicated by medium shade) or lower quartile (Q1 indicated by light shade). The colour coding clearly shows the multi-variate relationship between simulated N50, OA and the ox-VOC parameters. High values of OA may be associated with various combinations of ox-VOC parameters and with both high or low particle number concentrations. The figure is consistent with the challenge faced by state-of-the-art global models - despite simulating the particle number concentrations (N3 and N50) reasonably well models consistently under-predict OA mass concentrations (Kanakidou et al., 2005; Tsigaridis et al., 2014). The challenge in predicting both particle number concentrations and OA mass is re-visited in later sections.

N50 concentrations have a strong relationship with B_ELVOC up to about 5 times the ELVOC yield of 3.2%





(Kirkby et al., 2016) used in the model, above which there is more scatter in N50 caused by the other model processes and parameters. N50 is also related to B_LVOC, but with more scatter than for B_ELVOC. These re-

lationships show that N50 concentrations are strongly controlled in part by the production of B_ELVOC, which causes nucleation, and by the production of B_LVOC, which grows the nucleated clusters via kinetic condensation. There is no clear relationship between N50 and A_LVOC production. The likely reason for this is that anthropogenic VOCs are not spatially co-located with the biogenically-produced B_ELVOC, so there are fewer nuclei in polluted regions and hence much less effect of the A_LVOC on the growth of nuclei to larger sizes.

Including new, more accurate nucleation pathways into models is unlikely to improve the model performance with respect to N50 (a highly relevant model output for estimation of climate relevant aerosol-cloud interactions) unless the models also include adequate representation of B_LVOCs (Figure 7). Several studies have investigated nucleation capability and nucleation pathways of atmospheric molecules (Kulmala et al., 1998,

2004; Kirkby et al., 2011; Kurtén et al., 2008; Almeida et al., 2013; Riccobono et al., 2014; Kirkby et al., 2016) whereas the contribution of organic molecules to sub-3 nm cluster growth is relatively recent knowledge and the molecules involved are largely unidentified (Tröstl et al., 2016). The significance of B_LVOC is explored and established further in later sections.

OA concentrations in Figure 7 are found to be unrelated to B_ELVOC concentration showing that new particle formation has little effect on simulated OA mass in our model. Increases in all the other ox-VOC parameters generally increase OA, although there is a lot of scatter, particularly with the anthropogenic parameters, indi-

5  cating a strong multi-variate relationship for simulated OA. Global mean OA shows the strongest dependence on B_LVOC and B_SVOC_M and the highest global mean OA (darkest red in Figure 7) is simulated when B_LVOC and B_SVOC_M yields are more than 7.5 times the baseline yield of 13% (above 100% in Figure 7; producing over 113 Tg yr$^{-1}$ each).

10  We make two more observations from Figure 7. The simulated OA mass concentrations seem to have a stronger relationship with SVOCs than LVOCs, because SVOCs partition to larger particles which already hold substantial mass thereby having a greater impact on OA mass. Secondly OA concentrations appear to have a steeper in-





crease with increases in the biogenic ox-VOCs than their anthropogenic counterparts. This is because A_LVOCs or A_SVOCs grow fewer particles than their biogenic counterparts and as a result changes in their concentrations have a lesser impact on simulated OA mass. The involvement of anthropogenic precursors in particle formation and cluster growth is likely to change this picture (Molteni et al., 2018).

5  Overall we find that when particle number concentrations are low, the main difference between simulations that produce high amount of OA (simulations 16, 23, 35, 46 in Figure 4) and those that do not (see simulations 34, 55, 58 in Figure 4), is the relative concentrations of B_LVOC which grows freshly nucleated clusters before they can be scavenged by coagulation (see Figure 2 for parameter combinations). When particle number concentrations are high (due to high B_ELVOC or B_LVOC or both), the mass of OA produced is determined by the

10  combined effects of all other ox-VOCs. Parameter combinations such as in simulations 45 and 49 produce some of the highest global mean N3, N50 and OA within the ensemble. Parameter combinations when B_ELVOC and/or B_LVOC dominate significantly over the the rest of the ox-VOCs (such as simulations 8, 21, 41, 54 and 57) the increased competition between small particles for growth cause the available high volatility ox-VOCs to distribute on more particles, causing smaller increase in the particle mass.



**Figure 7.** Global annual mean N3 (top panel), N50 (middle panel) and OA (bottom panel) against the perturbed range of ox-VOCs, colour coded according to global mean OA concentrations: Q1 for OA values in the lower quartile, Q2 for the inter-quartile range and Q4 for OA values in the the upper quartile.





### 3.1.1   Ensemble comparison with observations — Structural deficiencies in the model

Figure 8 shows the monthly-mean timeseries of N3 at 34 ground sites. The 60-member ensemble is able to encompass the observations in all months at only 3 out of 34 sites. In most locations the annual mean model bias in each of the 60 ensemble members ranges between a factor of 3 underestimation to a factor of 2 overestimation, with underestimation being more prevalent. Riccobono et al. (2014), Dunne et al. (2016) and Gordon et al. (2016) have previously reported underestimation of winter-time particle number concentrations by the model. Here we find the winter-time underestimation continues even when B_ELVOC or B_LVOC ox-VOC parameters are at their highest settings. In addition to underestimation of modelled concentrations in winter, some ensemble members overestimate particle concentration in the summer (for example Aspvreten in Figure 8). These combined biases mean that the model overestimates the strength of the seasonal cycle compared to observations.

Figure 9 shows the monthly-mean timeseries of N50 at 31 ground sites. The 60-member ensemble is able to encompass the observations in all months at only 2 out of 31 sites. Like N3, the model bias for N50 in each of the 60 ensemble members range between a factor of 3 underestimation to a factor of 2 overestimation, with underestimation being more prevalent. The correlation coefficients (calculated for each ensemble member at each location using monthly mean simulated and observed N50 concentrations) in 22 out of 31 sites are higher than 0.5 (Figure not shown). The best correlation coefficients are observed in non-urban sites and the maximum underestimation and poorest correlation coefficients for N50 are observed at the polluted sites of Ispra and Marikana (Asmi et al., 2011; Vakkari et al., 2013). In contrast the ensemble performs significantly better at Hohenpeissenberg and Zugspitz, both high-altitude sites free from nearby anthropogenic influence only about 458 km from Ispra, and at Botsalano, representing a semi-clean environment, which is about 150 km from Marikana.

We also find that as the normalised mean bias factor (calculated between each simulation and observed values) reduces, the calculated correlation coefficient weakens. This implies that the model has a structural deficiency that cannot be resolved by perturbing the model parameters. The strong link between B_ELVOC and N3 indicates that the weakening of the correlation coefficient with the improvement of normalised mean bias factor is



related to nucleation: higher ELVOC production rates increase annual mean N3 concentrations, but the summer concentrations are affected much more than the winter concentrations, which weakens the correlation. We

suggest that a missing particle source such as anthropogenic pollutants − which are maximum in the winter due to low boundary layer height and increased local emissions from sources such as domestic heating − will rectify the model bias significantly. Alternatively, the poor model performance may be improved by exploring uncertainties in other parts of the model unrelated to SOA (Lee et al., 2013; Yoshioka et al., 2019).

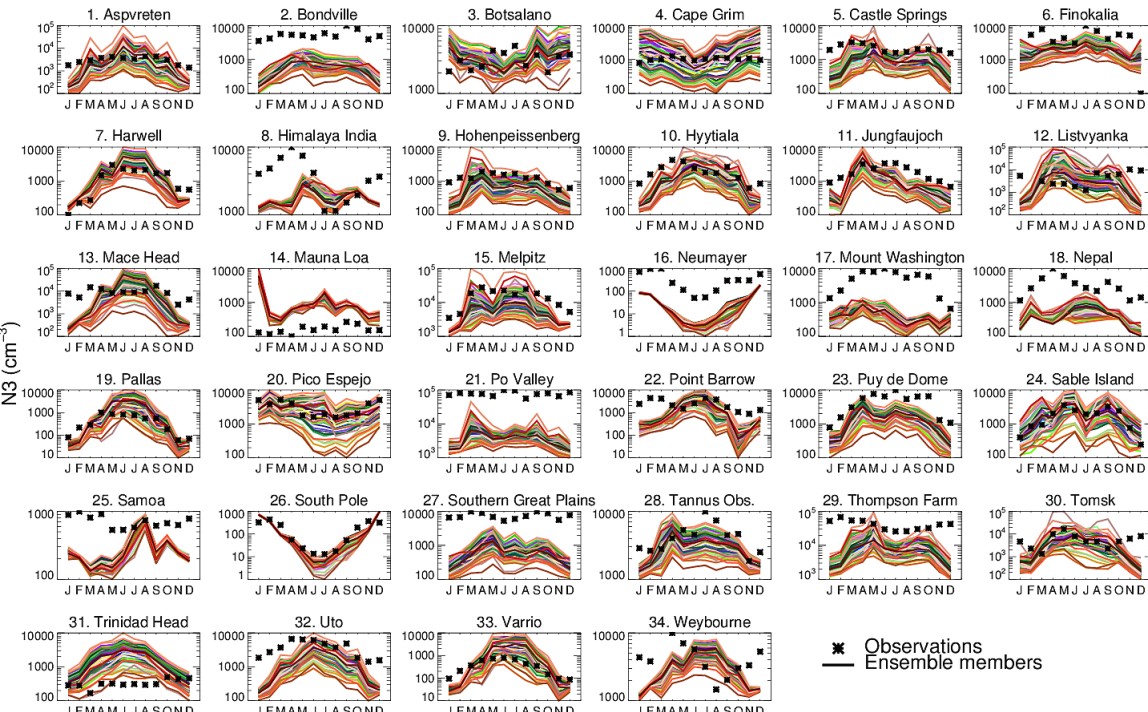

**Figure 8.** Seasonal cycle of simulated (solid coloured lines) and observed (black stars) monthly mean surface-level N3 concentrations at 34 ground-based sites. Each coloured line is an ensemble member.





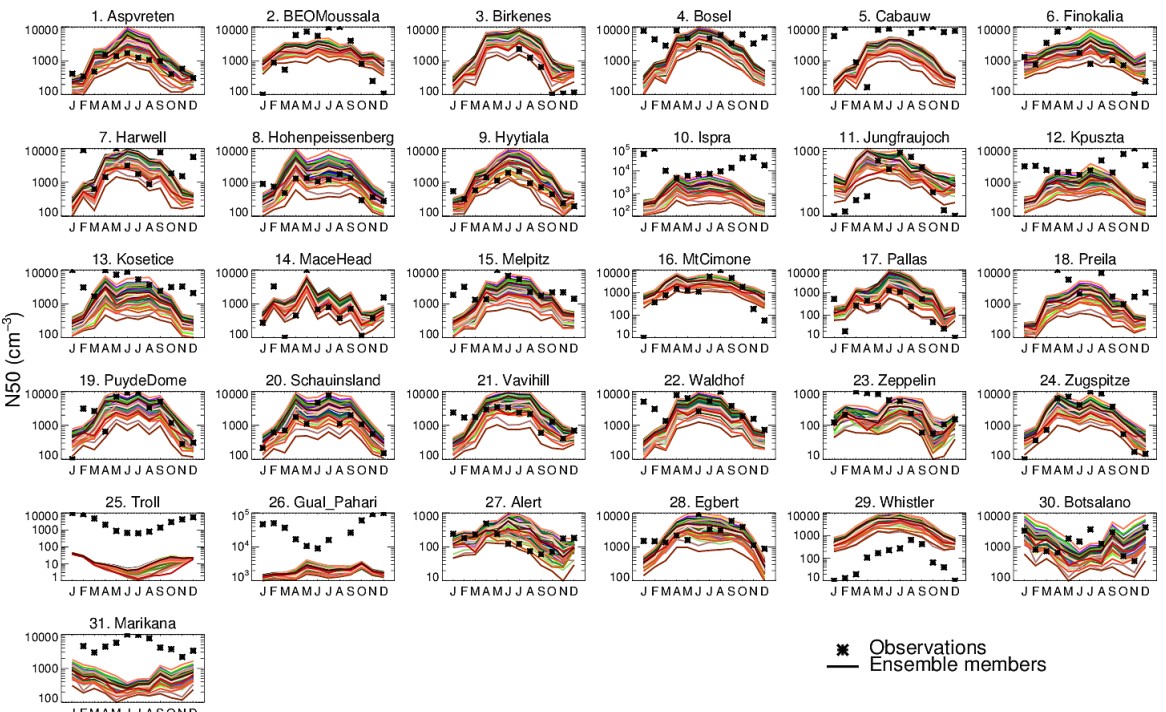

**Figure 9.** Seasonal cycle of simulated (solid coloured lines) and observed (black stars) monthly mean surface-level N50 concentrations at 31 ground-based sites. Each coloured line is an ensemble member.

## 3.2   Model skill across the 6-D parameter space

We now explore how the model skill in simulating observed N3, N50 and OA varies across the 6-D ox-VOC parameter space (Figures 10, 11 and 12). Because it is a 6-D space, it is important to note that for each sub-plot showing the relationship between two ox-VOCs, the other four parameters are varying randomly across each plane. A weak dependency between an ox-VOC and model skill does not imply that the contribution of the ox-VOC to OA and particle number concentration is unimportant. Rather, it implies that within the current
modelling framework its contribution can be compensated by changes in other ox-VOCs.





Some clear patterns in model skill are apparent across the 6 dimensions. N3 skill depends strongly on B_ELVOC, B_LVOC and A_LVOC (Figure 10). The skill is generally lower for B_ELVOC yields less than 6.4% from ozonolysis of $\alpha$-pinene (i.e.twice the baseline yield), irrespective of the value of other parameters (left column in Figure 10). The N3 skill is also generally low for values of A_LVOC greater than about 95 Tg yr$^{-1}$ (yield corresponding to 150% in Figure 10), although there are a few simulations that have reasonable skill above this value (second row from bottom). We note two regions in the 6-D space dominated by low model skill in N3 − where B_ELVOC yield is greater than 19.8% (6 times baseline) and B_LVOC is less than 113 Tg yr$^{-1}$ (corresponding to a yield of 100% in Figure 10 first column, second row from top) and where the sum of anthropogenic LVOC and SVOC is greater than 127 Tg yr$^{-1}$ (200% yield in Figure 10 fifth column, sixth row). There is also a general increase in skill for high values of B_LVOC (second column).

N50 skill has the strongest dependency on B_LVOC. The model is most skilful for B_LVOC greater than about 113 Tg yr$^{-1}$ (corresponding to a yield of 100%, Figure 11 second column) with a general increase in skill for higher values of B_LVOC. N50 skill is generally lower for B_ELVOC yields less than twice the baseline yield or 6.4%, although other parameter values and particularly high B_LVOC (Figure 11 first column, second row) improve model skill in some cases. The dependence of N50 skill on A_LVOC is much weaker than for N3, with high and low model skills spread across the entire parameter range. In contrast the N50 skill tends to be low for A_SVOCs greater than about 95 Tg yr$^{-1}$ (150% yield, Figure 11 bottom row).

Figures 10 and 11 show that model simulation of N3 and N50 is most skilful when B_ELVOC production is a factor of 2 to 8 higher than the baseline model B_ELVOC yields of 3.2% and 1.2% from O$_3$ and OH$^{.}$ oxidation reactions respectively (Kirkby et al., 2016). With less than 6.4% B_ELVOC yields from O$_3$ and OH$^{.}$ oxidation reactions, the model−observation match is consistently poor (shades of blue in Figure 10 or Figure 13). The best estimate of B_ELVOC yield to obtain reasonable agreement with observed N3, N50 and OA in our model (denoted by shades of red in Figure 13) is about a factor of 4 higher than the baseline B_ELVOC yield from ozonolysis of $\alpha$-pinene (Kirkby et al., 2016).





OA skill has the strongest joint dependency on B_LVOC and B_SVOC_M (Figure 12). The joint distribution suggests that the skill is poor if the sum of these SOA production rates exceeds about 226 Tg yr$^{-1}$ (up to 200% yield of both; Figure 12 second column, third row). However, for all other parameters there are skilful and unskilful simulations right across the 6-D parameter space.

Together, the variations in skill for N3, N50 and OA concentrations across the six dimensions show that the parameter space for a high N3 or N50 skill score does not overlap with the parameter space for a high OA skill score (Figures 10, 11 and 12). This gives an insight as to why models that are fine-tuned to simulate particle

number concentrations underestimate OA mass concentrations in the atmosphere − also identified as a persistent challenge for state-of-the-art global models (Kanakidou et al., 2005; Spracklen et al., 2011).

Our results reveal the problem of model equifinality, highlighted for the whole aerosol model by Lee et al. (2016). Equifinality means that there are multiple ways (i.e., parameter combinations) of achieving the same

model skill against observations, which makes it difficult to identify the best model (Beven, 2006). An important consequence of equifinality is that fine-tuning any one aspect of the model (for example, the nucleation mechanism) to achieve best model-observation agreement for any one variable (e.g., the particle number concentration) can be achieved with a wide range of settings of other parameters (e.g., parameters controlling overall OA mass production). While this may not affect the overall model skill in the particular evaluation, the

various parts of equally plausible parameter space may result in very different model behaviour in, say, climate projections.





**Figure 10.** Taylor Skill Score for model simulations against N3 observations across the 6-D parameter space. The x- and y-axes for a subplot show the total range of reaction yields (in%) over which each of the two parameters (as indicated by the plot labels at the top and right for each subplot respectively) is perturbed in the ensemble. Each triangle in a subplot represents a simulation and the color of the triangle indicates its Taylor skill score for N3. Darker shades of blue indicate low/poor Taylor skill score and darker shades of red represent high/good Taylor skill score. Figure A1 shows the same plot with ensemble members numbered.

*Note: Axis for B_ELVOC shows scaling factor for B_ELVOC yields. Axis for the rest show corresponding ox-VOC yields.*



**Figure 11.** Taylor Skill Score for model simulations against N50 observations across the 6-D parameter space. The x- and y-axes for a subplot show the total range of reaction yields (in%) over which each of the two parameters (as indicated by the plot labels at the top and right for each subplot respectively) is perturbed in the ensemble. Each triangle in a subplot represents a simulation and the color of the triangle indicates its Taylor skill score for N50. Darker shades of blue indicate low/poor Taylor skill score and darker shades of red represent high/good Taylor skill score. Figure A2 shows the same plot with ensemble members numbered.

*Note: Axis for B_ELVOC shows scaling factor for B_ELVOC yields. Axis for the rest show corresponding ox-VOC yields.*







**Figure 12.** Taylor Skill Score for model simulations against OA observations across the 6-D parameter space. The x- and y-axes for a subplot show the total range of reaction yields (in%) over which each of the two parameters (as indicated by the plot labels at the top and right for each subplot respectively) is perturbed in the ensemble. Each triangle in a subplot represents a simulation and the color of the triangle indicates its Taylor skill score for OA. Darker shades of blue indicate low/poor Taylor skill score and darker shades of red represent high/good Taylor skill score. Figure A3 shows the same plot with ensemble members numbered.

*Note: Axis for B_ELVOC shows scaling factor for B_ELVOC yields. Axis for the rest show corresponding ox-VOC yields.*





Figure 13 summarises Figures 10, 11 and 12 showing the model skill score in all three model outputs across the entire parameters space for all five ox-VOCs in 1-D. We find five simulations that are shaded red for all three model outputs across the 6 parameters: simulations 8, 17, 19, 21 and 41. The parameter combinations and the resulting Taylor Skill Score are listed in Tables 3 and A2.

| PPEM | B_ELVOC | B_LVOC | B_SVOC_M | B_SVOC_I | A_LVOC | A_LVOC | N3 | N50 | OA |
|------|---------|--------|----------|----------|--------|--------|-----|------|------|
|      | % yield $O_3$ (OH·) | | | Tg yr$^{-1}$ of SOA | | | | Taylor Skill Score | |
| 8  | 20.5 (7.7) | 122.72 | 59.76  | 31.89 | 43.55  | 64.96 | 0.23 | 0.10 | 0.15 |
| 17 | 9.7 (3.6)  | 178.15 | 74.25  | 49.27 | 75.89  | 73.15 | 0.23 | 0.10 | 0.12 |
| 19 | 13.8 (5.2) | 121.96 | 136.20 | 10.83 | 108.49 | 10.55 | 0.23 | 0.10 | 0.13 |
| 21 | 23.9 (8.9) | 226.26 | 42.2   | 4.65  | 96.24  | 87.28 | 0.22 | 0.11 | 0.13 |
| 41 | 13.2 (4.9) | 157.7  | 17.49  | 16.02 | 38.09  | 13.48 | 0.28 | 0.11 | 0.14 |

**Table 3.** Yield of B_ELVOC and Tg yr$^{-1}$ of SOA for other ox-VOCs with the corresponding Taylor skill scores for 5 ensemble members that are shaded red for all three outputs N3, N50 and OA in Figure 13. The parameter combinations used to produce the above ox-VOCs in the ensemble are listed in Table A2.

One ensemble member, simulation 41, scores reasonably well (Taylor Skill Scores of 0.28, 0.11 and 0.14 for N3, N50 and OA) in simulating observed mass and number of particles. It is the only simulation for which Taylor Skill Score in each of the three outputs are among the top 10 highest scores within the ensemble (see Table A1). The score is highest for N3, second highest for N50 (0.12 being the highest score) and the sixth highest for OA (0.19 being the highest score). For this simulation B_ELVOC is 4.1 times the baseline yield (i.e. about 13% B_ELVOC yield from $O_3$ and 5% yield from OH·), with about 157 Tg yr$^{-1}$ of SOA from B_LVOC, 33 Tg yr$^{-1}$ from B_SVOC (monoterpenes + isoprene), 38 Tg yr$^{-1}$ from A_LVOC and 13 Tg yr$^{-1}$ from A_SVOC. In this simulation the SOA production pathways are characterised by: i) high concentrations of both B_ELVOC and B_LVOC that ensure particle production via nucleation and subsequent growth of nucleated clusters; ii) relatively high A_LVOC concentrations that further helps to sustain the growth of small particles in the nucleation mode; iii) modest yields of SVOCs that ensure effective growth of particles up to CCN relevant sizes, improving skill scores for N50 and OA, while at the same time restraining the condensation sink and the loss of too many growing particles by coagulation scavenging. With Taylor Skill Scores of 0.28, 0.11 and 0.14 for N3, N50 and OA, simulation 41 has much scope for improvement. Nevertheless it exemplifies the characteristics required to improve the simulation of SOA in the model.





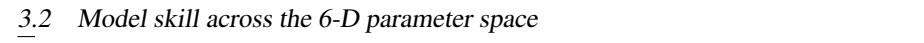

**Figure 13.** 1-D representation of the 6-D space filling design of experiments shaded by model performance. 18 subplots (labelled in plot) correspond to the design of experiments for six ox-VOCs, each shaded according to the Taylor skill score (TSS) in three model outputs (N3, N50, OA - in that order for each ox-VOC). Each subplot shows the parameter range perturbed for the yield (in %) of the corresponding ox-VOC* (x-axis). Each point on a subplot represents a simulation and the colour of the point indicates the performance of the simulation (TSS) against observed N3, N50 or OA. Darker shades of blue indicate low/poor TSS and darker shades of red represent high/good TSS. The plot identifies plausible and implausible parameter space for each ox-VOC. Simulations 8, 17, 19, 21 and 41 that are shaded red for all the three model outputs across the parameter space of all six ox-VOCs are labelled.

_Note: Axis for B_ELVOC shows scaling factor for B_ELVOC yields. Axis for the rest show corresponding ox-VOC yields._





## 4   Conclusions

We have used a perturbed parameter ensemble of 60 model simulations to explore how uncertainty in six biogenic and anthropogenic precursors affects organic aerosol mass and particle number concentrations. The ranges
for each parameter were chosen to encompass maximum uncertainty associated with organic compounds that affect three different stages of SOA formation − nucleation, cluster growth and particle growth. Simultaneous perturbations of the six parameters using a Latin Hypercube Sampling technique allow the effects of parameter-combinations rather than just individual parameters on model outputs to be explored. Three model outputs, the number concentration of particles larger than 3 nm diameter (N3), the number concentration of particles larger
than 50 nm diameter (N50), and the organic aerosol (OA) mass concentration, were compared against observations and the model skill score was then used to determine the skilful parts of parameter space.

The results expose a high degree of equifinality in the SOA model in which there are multiple ways of generating similar outputs (particle concentrations and OA mass). This is to be expected in a system with six free parameters and only three output variables of interest – that is, our six-component SOA model is underdetermined. Equifinality, or compensating parameter effects, limit the extent to which the best set of parameters can
5    be identified by comparing the simulations against observations. Our results suggest that the effects of three categories of volatile organic compounds can be detected by comparing the 60 simulations against observations: ELVOC, LVOC and SVOC. B_ELVOC is crucial for the formation of particles via nucleation. Thereafter contributions from LVOCs and SVOCs contribute to the growth of freshly-nucleated particles to produce a realistic N50 concentration and SOA mass.

B_ELVOC strongly influences model skill scores in N3 and to a lesser extent in N50. When B_ELVOC is low ($<$ twice the baseline yield of 3.2%), the ensemble consistently underestimates N3 and N50 number concentrations, irrespective of the availability of other ox-VOCs. We find the best model skills scores in N3, N50 and OA are achieved when the ELVOC yield from precursor VOCs is between 6−26%, with the most plausible
15    ELVOC yield estimate being aound 12.8%. Previously reported ELVOC yields from $\alpha$-pinene ozonolysis are at the lower end of this constrained range (3.2% with uncertainty range of +100%/-60% reported by Kirkby





et al. (2016), 7±4% reported by Ehn et al. (2014) and 3.4±1.7% reported by Jokinen et al. (2015). Ehn et al. (2014); Jokinen et al. (2015) report higher yields of ELVOC from other common biogenic VOCs and nucleating ELVOCs from anthropogenic sources have been reported by Molteni et al. (2018). Our range defines the plausible boundary for modelling nucleating ELVOCs by eliminating parameter space that results in low model skill scores. The best estimate for nucleating ELVOCs for the best model skill score within our plausible range is determined by the parameter combinations of other VOCs.

B_LVOC has the strongest influence on model skills in N3, N50 and OA. With insufficient B_LVOC, nucleated particles are lost before they can reach climate-relevant sizes in the model. High B_LVOC compensate to some extent for low B_ELVOC, depending on the value of other ox-VOCs. Both N50 and OA skill scores show the strongest relationship with B_LVOC but the parts of parameter space that favour high N50 skill and high OA skill contradict each other. Most skilful predictions of N50 tend to require over 113 Tg yr$^{-1}$ of B_LVOC, while the most skilful predictions of OA mass is favoured by B_LVOC lower than 113 Tg yr$^{-1}$. Our results show this inconsistency may be reconciled when B_LVOC greater than 113 Tg yr$^{-1}$ (favouring good N3 and N50 skill score) is accompanied by low B_SVOC_M such that the sum of B_LVOC + B_SVOC_M does not exceed about 226 Tg yr$^{-1}$ (favouring a good OA skill score, due to the joint dependency of OA skill score on these two ox-VOCs).

SVOCs (from monoterpene/isoprene/anthropogenic sources) are important to simulate realistic number concentrations of climate-relevant sized particles such as N50 and OA mass concentrations. SVOCs generated from $\alpha$-pinene in the model are spatially co-located with nucleated particles and hence have a strong effect on their growth to larger sizes. OA skills in particular show a strong dependence on SVOC_M as discussed above.

We cannot determine the plausible or implausible parameter space for those ox-VOC parameters which show a weak relationship with the model skill score. Model skill in N3, N50 and OA have the weakest relationship with B_SVOC_I. This is because in the current model setup the role of B_SVOC_I in growing particles by mass-based partitioning can be easily compensated for by other oxidised VOCs. The relationship between anthropogenic oxidised VOCs and model skill score in OA are also weak, although A_LVOC shows a strong re-





lationship with N3 skill score and A_SVOC with N50 skill score. The weak relationship between anthropogenic SVOCs with simulated OA mass is because the clusters produced by biogenic nucleation are not spatially co-localated with the anthropogenic LVOCs or SVOCs. We expect the relationship between anthropogenic VOCs and model skill scores in N3, N50 and OA to strengthen when the role of anthropogenic oxidised VOCs in nucleation and cluster growth Molteni et al. (2018) are represented.

Our results point to a structural deficiency in the model. The perturbed parameter ensemble tends to exaggerate the observed seasonal cycle of particle concentrations, overestimating summer and underestimating winter-time

particle concentrations. We suggest this is due to SOA sources in models being predominantly biogenic. Increases in the concentration of nucleating biogenic aerosols or in the complexities of nucleation mechanisms will not improve the model's ability to replicate the observed seasonal cycle. We expect the inclusion of anthropogenic sources in nucleation and in initial cluster growth for SOA particles in models would significantly improve the simulated aerosol seasonal cycle. It is important to explore the anthropogenic contribution to cluster-

growth because such pathways could considerably impact model estimates of anthropogenic aerosol forcing by differentially affecting the present-day and pre-industrial atmospheres (Carslaw et al., 2013).

*Data availability.* Measurement data, model data used in this paper can be made available upon request from the corresponding author.




# 1   Statistical summary

| PPEM | TSS N3 | TSS N50 | TSS OA | | NMBF N3 | NMBF N50 | NMBF OA | | CORR N3 | CORR N50 | CORR OA |
|---|---|---|---|---|---|---|---|---|---|---|---|
| 1 | 0.23 | 0.10 | 0.07 | | -0.49 | -0.55 | 2.00 | | 0.48 | 0.49 | 0.20 |
| 2 | 0.20 | 0.09 | 0.11 | | -0.72 | -0.71 | 1.67 | | 0.49 | 0.51 | 0.24 |
| 3 | 0.25 | 0.10 | 0.08 | | -0.48 | -0.58 | 1.62 | | 0.50 | 0.49 | 0.19 |
| 4 | 0.21 | 0.10 | 0.07 | | -0.56 | -0.60 | 2.36 | | 0.46 | 0.49 | 0.21 |
| 5 | 0.23 | 0.10 | 0.05 | | -0.43 | -0.52 | 2.37 | | 0.47 | 0.48 | 0.19 |
| 6 | 0.21 | 0.10 | 0.10 | | -0.20 | -0.43 | 1.63 | | 0.37 | 0.44 | 0.23 |
| 7 | 0.19 | 0.08 | 0.14 | | -0.89 | -0.89 | 1.35 | | 0.49 | 0.54 | 0.27 |
| 8 | 0.23 | 0.10 | 0.15 | | -0.19 | -0.52 | 0.71 | | 0.40 | 0.45 | 0.25 |
| 9 | 0.20 | 0.09 | 0.12 | | -0.40 | -0.73 | 1.38 | | 0.37 | 0.50 | 0.24 |
| 10 | 0.22 | 0.10 | 0.06 | | -0.57 | -0.57 | 2.20 | | 0.49 | 0.50 | 0.19 |
| 11 | 0.19 | 0.09 | 0.10 | | -0.40 | -0.58 | 2.18 | | 0.36 | 0.46 | 0.25 |
| 12 | 0.20 | 0.09 | 0.14 | | -0.70 | -0.85 | 1.02 | | 0.43 | 0.53 | 0.25 |
| 13 | 0.20 | 0.09 | 0.06 | | -0.73 | -0.70 | 2.19 | | 0.50 | 0.52 | 0.20 |
| 14 | 0.23 | 0.09 | 0.09 | | -0.55 | -0.69 | 1.65 | | 0.48 | 0.50 | 0.22 |
| 15 | 0.19 | 0.09 | 0.11 | | -0.27 | -0.62 | 1.27 | | 0.33 | 0.49 | 0.22 |
| 16 | 0.16 | 0.10 | 0.06 | | -1.28 | -0.92 | 2.48 | | 0.60 | 0.60 | 0.19 |
| 17 | 0.23 | 0.10 | 0.12 | | -0.59 | -0.62 | 1.33 | | 0.50 | 0.49 | 0.23 |
| 18 | 0.20 | 0.09 | 0.09 | | -0.47 | -0.65 | 1.76 | | 0.40 | 0.50 | 0.21 |
| 19 | 0.23 | 0.10 | 0.13 | | -0.45 | -0.57 | 0.95 | | 0.45 | 0.48 | 0.23 |
| 20 | 0.19 | 0.09 | 0.12 | | -0.22 | -0.59 | 1.41 | | 0.33 | 0.46 | 0.24 |
| 21 | 0.22 | 0.11 | 0.13 | | -0.23 | -0.39 | 1.27 | | 0.40 | 0.43 | 0.24 |
| 22 | 0.21 | 0.10 | 0.06 | | -0.28 | -0.44 | 2.50 | | 0.39 | 0.45 | 0.20 |
| 23 | 0.15 | 0.09 | 0.08 | | -1.44 | -1.04 | 2.26 | | 0.61 | 0.61 | 0.22 |
| 24 | 0.21 | 0.09 | 0.11 | | -0.73 | -0.71 | 1.63 | | 0.51 | 0.51 | 0.23 |
| 25 | 0.20 | 0.10 | 0.09 | | -0.49 | -0.58 | 1.94 | | 0.42 | 0.48 | 0.23 |
| 26 | 0.26 | 0.11 | 0.05 | | -0.44 | -0.52 | 2.29 | | 0.52 | 0.49 | 0.17 |
| 27 | 0.20 | 0.09 | 0.14 | | -0.20 | -0.65 | 0.75 | | 0.33 | 0.48 | 0.24 |
| 28 | 0.26 | 0.10 | 0.11 | | -0.44 | -0.55 | 1.24 | | 0.51 | 0.47 | 0.21 |
| 29 | 0.23 | 0.09 | 0.06 | | -0.33 | -0.62 | 2.25 | | 0.43 | 0.47 | 0.20 |
| 30 | 0.21 | 0.09 | 0.14 | | -0.40 | -0.60 | 1.37 | | 0.41 | 0.46 | 0.27 |
| 31 | 0.22 | 0.10 | 0.07 | | -0.43 | -0.49 | 2.08 | | 0.45 | 0.48 | 0.19 |
| 32 | 0.25 | 0.11 | 0.09 | | -0.19 | -0.36 | 1.54 | | 0.44 | 0.43 | 0.20 |
| 33 | 0.18 | 0.09 | 0.09 | | -1.02 | -0.92 | 2.09 | | 0.53 | 0.55 | 0.24 |
| 34 | 0.21 | 0.09 | 0.12 | | -0.92 | -0.91 | 0.98 | | 0.55 | 0.56 | 0.22 |
| 35 | 0.18 | 0.09 | 0.06 | | -0.96 | -0.80 | 2.60 | | 0.53 | 0.55 | 0.21 |
| 36 | 0.19 | 0.09 | 0.16 | | -0.33 | -0.56 | 1.30 | | 0.34 | 0.45 | 0.29 |
| 37 | 0.22 | 0.10 | 0.10 | | -0.56 | -0.54 | 1.64 | | 0.49 | 0.49 | 0.22 |
| 38 | 0.20 | 0.08 | 0.09 | | -0.55 | -0.96 | 1.34 | | 0.37 | 0.58 | 0.20 |
| 39 | 0.17 | 0.08 | 0.18 | | -0.80 | -1.04 | 0.65 | | 0.34 | 0.59 | 0.30 |
| 40 | 0.15 | 0.08 | 0.16 | | -1.18 | -1.11 | 1.55 | | 0.49 | 0.57 | 0.30 |
| 41 | 0.28 | 0.11 | 0.14 | | -0.21 | -0.49 | 0.28 | | 0.47 | 0.46 | 0.23 |
| 42 | 0.17 | 0.08 | 0.08 | | -1.23 | -1.12 | 2.03 | | 0.59 | 0.59 | 0.21 |
| 43 | 0.23 | 0.10 | 0.07 | | -0.27 | -0.43 | 2.03 | | 0.42 | 0.45 | 0.20 |
| 44 | 0.18 | 0.09 | 0.14 | | -1.10 | -0.86 | 1.58 | | 0.55 | 0.55 | 0.27 |
| 45 | 0.18 | 0.09 | 0.07 | | -0.35 | -0.55 | 2.34 | | 0.34 | 0.47 | 0.22 |
| 46 | 0.22 | 0.10 | 0.06 | | -0.86 | -0.72 | 1.99 | | 0.58 | 0.55 | 0.19 |
| 47 | 0.20 | 0.07 | 0.17 | | -0.50 | -1.03 | 0.13 | | 0.35 | 0.54 | 0.36 |
| 48 | 0.22 | 0.11 | 0.05 | | -0.33 | -0.40 | 2.52 | | 0.42 | 0.45 | 0.20 |
| 49 | 0.25 | 0.11 | 0.04 | | -0.22 | -0.46 | 2.34 | | 0.44 | 0.47 | 0.17 |
| 50 | 0.16 | 0.07 | 0.17 | | -0.72 | -1.05 | 1.31 | | 0.30 | 0.56 | 0.31 |
| 51 | 0.15 | 0.07 | 0.10 | | -0.35 | -1.40 | 1.52 | | 0.27 | 0.64 | 0.22 |
| 52 | 0.16 | 0.09 | 0.08 | | -1.10 | -0.98 | 1.91 | | 0.52 | 0.58 | 0.21 |
| 53 | 0.14 | 0.09 | 0.14 | | -1.62 | -1.14 | 1.79 | | 0.65 | 0.64 | 0.19 |
| 54 | 0.19 | 0.09 | 0.14 | | 0.05 | -0.65 | 0.44 | | 0.35 | 0.46 | 0.24 |
| 55 | 0.12 | 0.07 | 0.14 | | -1.91 | -1.82 | 1.02 | | 0.63 | 0.70 | 0.26 |
| 56 | 0.17 | 0.09 | 0.12 | | -0.55 | -0.74/ | 1.77 | | 0.35 | 0.50 | 0.26 |
| 57 | 0.23 | 0.12 | 0.11 | | -0.09 | -0.27 | 1.14 | | 0.40 | 0.42 | 0.20 |
| 58 | 0.15 | 0.08 | 0.19 | | -1.38 | -1.32 | 0.24 | | 0.55 | 0.62 | 0.36 |
| 59 | 0.22 | 0.09 | 0.09 | | -0.25 | -0.89 | 0.78 | | 0.36 | 0.57 | 0.17 |
| 60 | 0.21 | 0.10 | 0.05 | | -0.58 | -0.56 | 3.04 | | 0.47 | 0.49 | 0.21 |
| | | | | | | | | | | | |
| MIN | 0.12 | 0.07 | 0.04 | 0.00 | -1.91 | -1.82 | 0.13 | 0.00 | 0.27 | 0.42 | 0.17 |
| MAX | 0.28 | 0.12 | 0.19 | 0.05 | -0.27 | 3.04 | 0.00 | 0.65 | 0.70 | 0.36 | |

**Table A1.** Summary statistics - Taylor Skill Score (TSS), Normalised Mean Bias Factor (NMBF) and Pearson correlation coefficient (R) - for each ensemble member, based on model comparison against observations of N3, N50 and OA. The 10 best and 10 worst simulations in each category are highlighted in red and blue respectively.





| PPEM | B_ELVOC | B_LVOC | B_SVOC_M | B_SVOC_I | A_LVOC | A_LVOC | N3 | N50 | OA |
|---|---|---|---|---|---|---|---|---|---|
|  | Scaling factor for yield | % yield | | | | | Taylor Skill Score | | |
| 8 | 6.4 | 108.75 | 52.96 | 8.3 | 68.8 | 102.34 | 0.23 | 0.10 | 0.15 |
| 17 | 3.04 | 157.88 | 65.8 | 12.82 | 119.9 | 115.24 | 0.23 | 0.10 | 0.12 |
| 19 | 4.32 | 108.08 | 120.7 | 2.82 | 171.5 | 16.63 | 0.23 | 0.10 | 0.13 |
| 21 | 7.49 | 200.5 | 37.46 | 1.21 | 152.03 | 137.5 | 0.22 | 0.11 | 0.13 |
| 41 | 4.14 | 139.75 | 15.58 | 4.17 | 60.18 | 21.25 | 0.28 | 0.11 | 0.14 |

**Table A2.** Scaling factor for B_ELVOC and yields of other ox-VOCs with the corresponding Taylor skill scores for 5 ensemble members that are shaded red for all three outputs N3, N50 and OA in Figure 13.

*1 Statistical summary*                                                                                                5

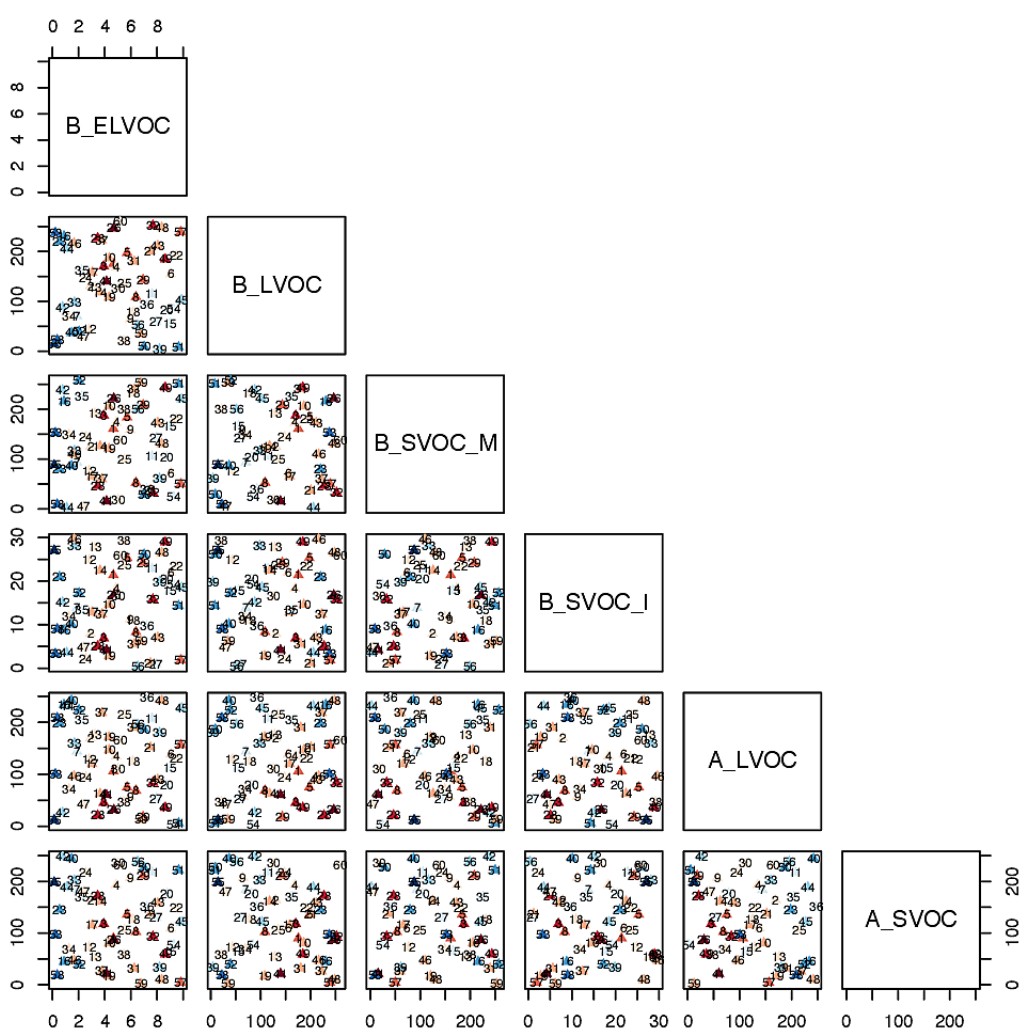

**Figure A1.** Taylor Skill Score for model simulations against N3 observations across the 6-D parameter space. The x- and y-axes for a subplot show the total range of reaction yields (in%) over which each of the two parameters (as indicated by the plot labels at the top and right for each subplot respectively) is perturbed in the ensemble. Each triangle in a subplot represents a simulation (labelled 1-60) and the color of the triangle indicates its Taylor skill score for N3. Darker shades of blue indicate low/poor Taylor skill score and darker shades of red represent high/good Taylor skill score.

*Note: Axis for B_ELVOC shows scaling factor for B_ELVOC yields. Axis for the rest show corresponding ox-VOC yields.*





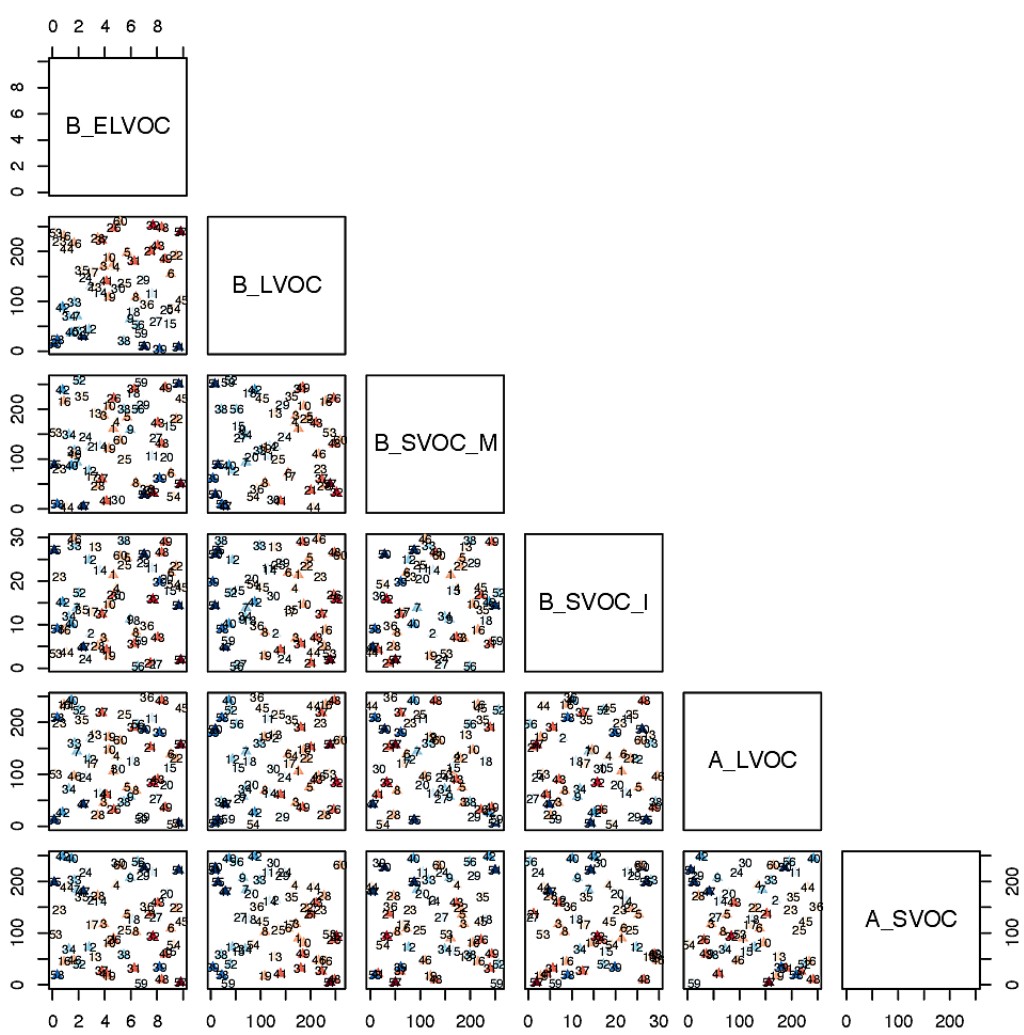

**Figure A2.** Taylor Skill Score for model simulations against N50 observations across the 6-D parameter space. The x- and y-axes for a subplot show the total range of reaction yields (in%) over which each of the two parameters (as indicated by the plot labels at the top and right for each subplot respectively) is perturbed in the ensemble. Each triangle in a subplot represents a simulation (labelled 1-60) and the color of the triangle indicates its Taylor skill score for N50. Darker shades of blue indicate low/poor Taylor skill score and darker shades of red represent high/good Taylor skill score.

*Note: Axis for B_ELVOC shows scaling factor for B_ELVOC yields. Axis for the rest show corresponding ox-VOC yields.*





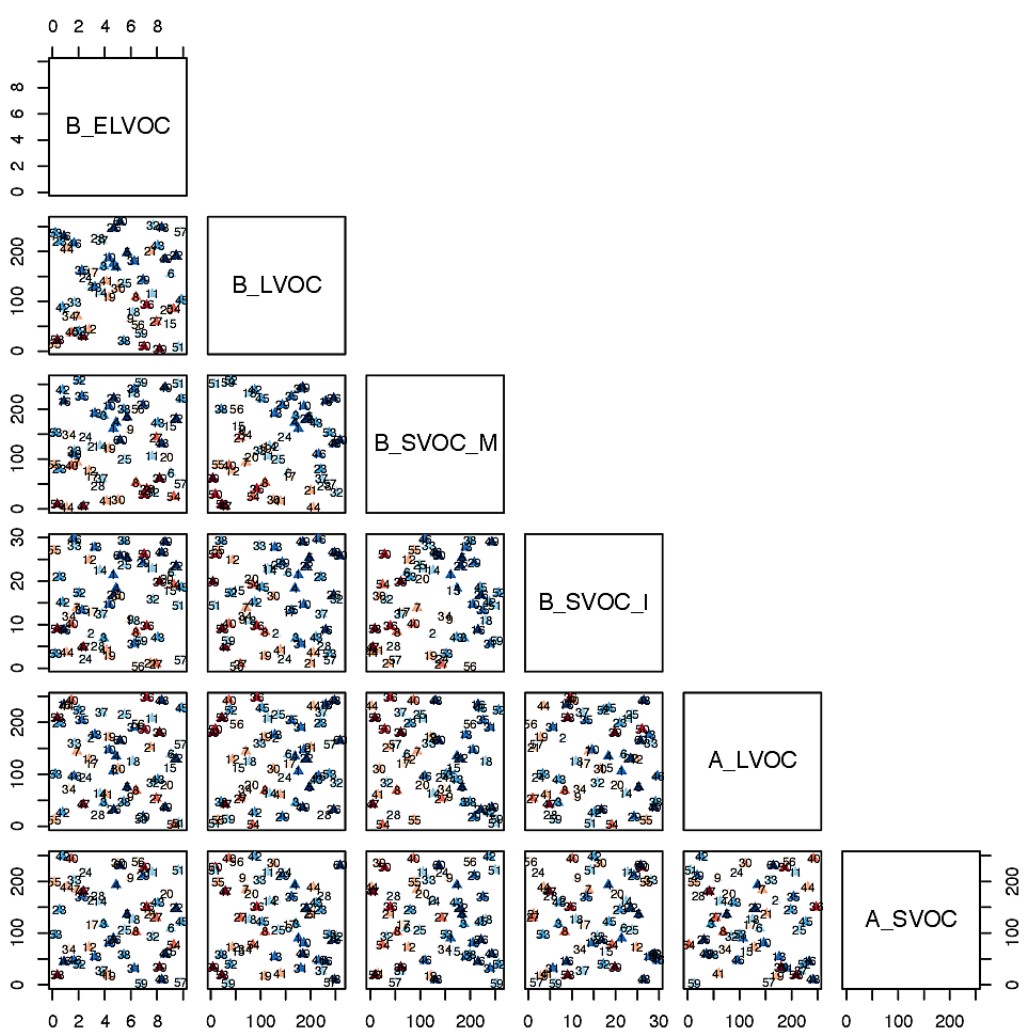

**Figure A3.** Taylor Skill Score for model simulations against OA observations across the 6-D parameter space. The x- and y-axes for a subplot show the total range of reaction yields (in%) over which each of the two parameters (as indicated by the plot labels at the top and right for each subplot respectively) is perturbed in the ensemble. Each triangle in a subplot represents a simulation (labelled 1-60) and the color of the triangle indicates its Taylor skill score for OA. Darker shades of blue indicate low/poor Taylor skill score and darker shades of red represent high/good Taylor skill score.

*Note: Axis for B_ELVOC shows scaling factor for B_ELVOC yields. Axis for the rest show corresponding ox-VOC yields.*



*Author contributions.* KS, KSC and KP designed research and interpreted the results. KP helped with technical aspects of setting up the model to run the ensemble. JSJ advised on designing the perturbed parameter ensemble. KS ran the

perturbed parameter ensemble and analysed the data. CLR collated the OA data and JB collated the N50 data used for model-observation comparison. CES provided code to represent the mass-based partitioning in the model. KS and KSC wrote the article, with comments from other co-authors.

*Competing interests.* The authors declare that they have no conflict of interest.

*Acknowledgements.* K. Sengupta was funded through the EC Seventh Framework Programme (Marie Curie Initial Training Network 'CLOUD-TRAIN' (no. 316662)). This work was undertaken on ARC1, part of the High Performance Computing facilities at the University of Leeds, UK. This research was supported by the Natural Environment Research Council (NERC) under Grants NE/J024252/1, NE/I020059/1 and NE/S015396/1. We would like to thank Sarah Monks for the monthly mean

Monoterpene and isoprene data used in this study. For the contribution of ground station data, we would like to thank: Dr R. Leaitch at Environment Canada (Whistler, Egbert and Alert stations), Dr V. Vakkari and Dr L. Laakso at the University of Helsinki (Botsalano and Welegund stations), Dr T. Laurila (Tiksi station) and Dr A. Hyvarinen (Gual Pahari station) at the Finnish Meteorological institute, Dr C. Lunder and Dr M. Fiebig (Troll station) at the Norwegian institute for Air research, and Dr Ari Asmi at the University of Helsinki for the provision of European N50 data. Ground station observations

were collated via GASSP and public data on the EBAS database (http://ebas.nilu.no). The EBAS database has largely been funded by the UN-ECE CLRTAP (EMEP), AMAP and through NILU internal resources. Specific developments have been possible due to projects like EUSAAR (EU-FP5)(EBAS web interface), EBAS-Online (Norwegian Research Council INFRA) (upgrading of database platform) and HTAP (European Commission DG-ENV)(import and export routines to build a secondary repository in support of www.htap.org). A large number of specific projects have supported development of

data and meta data reporting schemes in dialogue with data providers (EU)(CREATE, ACTRIS and others). For a complete list of programmes and projects for which EBAS serves as a database, please consult the information box in the Framework filter of the web interface. These are all highly acknowledged for their support.





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
