# Peer review of "A global model perturbed parameter ensemble study of secondary organic aerosol formation"

_Atmospheric Chemistry and Physics, 2020_

## Referee Comment (RC2)

The modeling of secondary organic aerosol (SOA) formation involves enormous complexity. Sengupta et al. used a perturbed parameter ensemble of 60 model simulations to evaluate six representative SOA precursors on aerosol number concentration and organic aerosol mass. The model results have been evaluated against measurements from sites across the globe. The authors determined the critical role of ELVOC and LVOC in new particle formation and growth, and the importance of LVOC and SVOC in simulating organic mass. They also found a high degree of equifinality in the SOA model. Parameter combinations that are best for aerosol number concentration are worst for organic aerosol mass. This manuscript provides a novel approach to evaluate effects of parameter combinations, rather than individual parameters, on both aerosol number and mass concentration that matter for climate impacts. I would recommend publication of this manuscript after addressing minor issues below.

1. A brief introduction on the meaning and climate implications of N3 and N50 should be added somewhere in the introduction (not later in Page 20).

2. Some more information about the GLOMAP model is needed in Section 2.1, e.g. what are the emission inventories used here? How does the model treat the removal of aerosols? What do anthropogenic VOCs include?

3. Table 2: B_ELVOC does not produce any SOA in the default scheme? Why?

4. Figure 2: Would you please say a bit more about how you design these 60 experiments? It may be clearer to include an appendix table to show the combination of yields you chose and explain why. At P14 L18 you said, "the total global production of SOA varies by only a factor of 4 and the lowest value is 220 Tg yr−1". This really depends on the choice of yields combination, right? If you choose 0 for all yields, the SOA would be zero? And from Figure 2, I did not see an experiment that uses maximum scaling/yields for all sources. Would such an experiment show a much higher SOA production than 850 Tg/yr?

5. P13: Could you please provide more information on TSS? How did you calculate R and Ro? Is the normalized standard deviation calculated by all model vs observation monthly mean data across the globe? And could you give a more intuitive interpretation of the TSS values? Later in Figure 10-12 you defined "low/poor" versus "high/good" TSS, which should be explained here as well.

6. P14 L4: "The global mass of SOA produced in the model simulations ranges from 220 to 850 Tg yr−1". Doesn't this include the 107 Tg/yr of SOA produced by the default SOA scheme (based on Table 2)?

7. P19 L18: Why do high B_SVOC_I and A_LVOC suppress particle growth?

8. Caption of Figure 7 is wrong. N3 is not shown. Q2 and IQR not consistent.

9. Figure 7: Could the "global mean" OA be as high as 3-10 ug/m3? Did you do an area-averaged OA over the whole globe (including ocean), or the total land area, or averaged over the available sites?

10. P20 L13-14 (the first sentence actually) should refer to Figure 7.

11. P20 L21-22 Did Figure 7 include any information of nucleation? "Including new, more accurate nucleation pathways into models is unlikely to improve the model performance with respect to N50." This sentence is confusing.

12. P20 The whole paragraph "OA concentrations in Figure 7 are found to be unrelated to B_ELVOC…" It could be helpful to include regression lines and some statistics (e.g. slope, significance, correlations) to support your statements here.

13. P21 "This is because A_LVOCs or A_SVOCs grow fewer particles than their biogenic counterparts and as a result changes in their concentrations have a lesser impact on simulated OA mass. " Again, is such information included in Figure 7? Is it possible that it's just because anthropogenic sources produce much less SOA than biogenic sources on the global scale?

14. P23 L24 "the model overestimates the strength of seasonal cycle". This is only true for just a few sites, right?

15. Why not include color bars to show the TSS range in Figure 10-12?

16. P26 L17-26. These interpretations are very confusing, and Figure 10-12 are hard to understand. For example:

"where B_ELVOC yield is greater than 19.8%": Does this refer to first column in Figure 10?

"B_LVOC is less than 113 Tg yr−1 (corresponding to a yield of 100% in Figure 10 first column, second row from top)": I think this should refer to second column in Figure 10? The "first column, second row from top" refers to the scatter plot with B_ELVOC on the x-axis and B_SVOC_M on the y-axis. I don't think this is the one you are referring to here.

"where the sum of anthropogenic LVOC and SVOC is greater than 127 Tg yr$^{-1}$ (200% yield in Figure 10 fifth column, sixth row)." – There is no sixth row in Figure 10. Should be fifth row?

17. P26 Second paragraph about N50. Again, "Figure 11 first column, second row" refers to the scatter plot with B_ELVOC on the x-axis and B_SVOC_M on the y-axis, not consistent with what you are saying here.

18. Figure 10-12 may need to be rearranged for an easier interpretation. For example, both X and Y axis should include 6 ox-VOCs and in the same order, i.e. have B_ELVOC, B_LVOC, … A_SVOC on the X-axis from left to right, and on the Y-axis from bottom to top. Also please define "good" and "bad" TSS scores.

19. P31 Second line "… for all five ox-VOCs in 1-D": should be "six ox-VOCs" here.

20. P34 I like the statements about B_LVOC, B_SVOC_I/M, A_SVOC which are much clearer than in the results section, but would you please refer back to the figures from which you draw those conclusions? In general Section 3.2 is hard to follow with quite a few typos/errors and Figure 10-12 are not easy to interpret. Please check through the manuscript for potential mismatches and typos.

---

## Referee Comment (RC1) · Anonymous Referee #1 · 13 Sep 2020

This manuscript explores the impacts of six yield parameters on particle number concentration and organic aerosol concentration based on 60 sensitivity simulations in global models. The six parameters include the yields of ELVOC/LVOC/SVOC from the oxidation of isoprene, monoterpenes, and anthropogenic VOCs. The simulated concentration of particles >3nm (N3), particles > 50nm (N50), and OA are extensively compared against measurements around the world. The manuscript discussed critical compensating parameter effects, which limit our ability to retrieve best set of parameters by comparing model and measurements. Further, it is found that parameters leading to best simulation of N3 and N50 are the worst of OA concentration, because these three attributes are driven by OVOCs with different volatilities. It is delightful to read the manuscript. The authors do a terrific job in clearly commuting and visualiz-

ing the results from 60 sensitivity simulations. Overall, the manuscript has immediate impacts on the atmospheric chemistry community and nicely fits in the scope of ACP. I recommend publication after minor revision.

Comments 1. As clearly demonstrated in the manuscript, it is challenging to retrieve best set of parameters by comparing model and measurements. To some extent, this emphasizes the importance of provide accurate parameters based on laboratory experiments. It would be great if the authors could provide some suggestions to experimentalists. 2. One clarification question: does N3 refer to particles larger or smaller than 3nm? If it is larger than 3nm as defined in Page 4 Line 20, is N50 part of N3? 3. Page 14 Line 24. Based on the index in figure 4, simulation 9 should be subplot (4,3), instead of subplot (3,4). 4. Figures 4-6. I wonder if it is better to organize the subplots in the same order as figure 2, which will facilitate locating the simulations that are discussed in the manuscript. Just a thought. 5. Figure 7. It is "Q2" in the caption, but "IQR" in the legend. Please be consistent. Also, "N3" is mentioned in the caption, but not included in the figure. Please add N3 plots to the figure to provide a complete picture. 6. Please elaborate the discussions in section 3.2, as it is not straightforward how to read figures 10-12.

---

## Author Comment (AC1) · 18 Nov 2020

ACPD Interactive comment Printer-friendly version Discussion paper Atmos. Chem. Phys. Discuss., https://doi.org/10.5194/acp-2020-756-RC1, 2020 © Author(s) 2020. This work is distributed under the Creative Commons Attribution 4.0 License. Interactive comment on "A global model perturbed parameter ensemble study of secondary organic aerosol formation" by Kamalika Sengupta et al. Anonymous Referee #1 This manuscript explores the impacts of six yield parameters on particle number concentration and organic aerosol concentration based on 60 sensitivity simulations in global models. The six parameters include the yields of ELVOC/LVOC/SVOC from the oxidation of isoprene, monoterpenes, and anthropogenic VOCs. The simulated concentration of particles >3nm (N3), particles > 50nm (N50), and OA are extensively compared against measurements around the world. The manuscript discussed critical compensating parameter effects, which limit our ability to retrieve best set of parameters by comparing model and measurements. Further, it is found that parameters leading to best simulation of N3 and N50 are the worst of OA concentration, because these three attributes are driven by OVOCs with different volatilities. It is delightful to read the manuscript. The authors do a terrific job in clearly commuting and visualizC1 ACPD Interactive comment Printer-friendly version Discussion paper ing the results from 60 sensitivity simulations. Overall, the manuscript has immediate impacts on the atmospheric chemistry community and nicely fits in the scope of ACP. I recommend publication after minor revision.

Thank you for the constructive feedback. Please see our replies inline. Any text added in the revised manuscript is shown below in **bold, italics**.

Comments 1. As clearly demonstrated in the manuscript, it is challenging to retrieve best set of parameters by comparing model and measurements. To some extent, this emphasizes the importance of provide accurate parameters based on laboratory experiments. It would be great if the authors could provide some suggestions to experimentalists.

Suggested ranges are discussed in the Results and Conclusion sections. For example in P35 L24:

***We find the best model skills scores in N3, N50 and OA are achieved when the ELVOC yield from precursor VOCs is between 6-26%, with the most plausible ELVOC yield estimate being around 12.8%.***

2. One clarification question: does N3 refer to particles larger or smaller than 3nm? If it is larger than 3nm as defined in Page 4 Line 20, is N50 part of N3?

N3 refers to particles larger than 3nm.

Yes, N50 is part of N3. More information on N3 and N50 has been added in Section 2.3, in the revised manuscript.

3. Page 14 Line 24. Based on the index in figure 4, simulation 9 should be subplot (4,3), instead of subplot (3,4).

Corrected.

4. Figures 4-6. I wonder if it is better to organize the subplots in the same order as figure 2, which will facilitate locating the simulations that are discussed in the manuscript. Just a thought.

Currently Figure 4 is arranged in order of increasing global mean OA. This order makes it easy to see how simulations that have very close global mean values of OA, have widely different regional distributions (simulations 9 and 36, lines 11-20 in P15).

Figures 5 and 6 are both arranged in increasing order of global mean N3. This order makes it easy to spot simulations in which the particle growth from 3nm to 50nm has been affected by the parameter combinations (P20 L2-10). Such simulations are easy to spot in Figure 6 because of the sudden blue plots appearing amidst increasingly red ones.

Ordering the subplots in Figures 4-6 from 1-60 (as in Figure 2) will make it harder to spot the above patterns. To help the reader locate the simulations, we have the subplot numbering system.

Based on the above we think it is best to keep the order of subplots in Figures 4-6 unchanged.

5. Figure 7. It is "Q2" in the caption, but "IQR" in the legend. Please be consistent. Also, "N3" is mentioned in the caption, but not included in the figure. Please add N3 plots to the figure to provide a complete picture.

Corrected.

6. Please elaborate the discussions in section 3.2, as it is not straightforward how to read figures 10-12. Interactive comment on Atmos. Chem. Phys. Discuss., https://doi.org/10.5194/acp-2020-756, 2020. C2

The following text on how to interpret Figures 10-12 is added P26 L6 in the revised manuscript.

*Each scatter plot in Figures 10, 11 and 12 show the relationship between two ox-VOCs for each of the sixty ensemble members. Note the values for B_ELVOC yields shown in Figures 10, 11 and 12 and Figures A3, A4 and A5 are scaling factors which have to be multiplied to the baseline model yields (Table 2) to get the B_ELVOC*

*yields for the ensemble members. For the rest of the ox-VOCs the values shown are yield values in % which can be converted to Tg yr⁻¹ using Table 2.* Because it is a 6-D space, it is also important to note that the other four parameters are varying randomly across each plane.

*We use Figures 10, 11 and 12 to identify patterns of dependencies of the Taylor skill scores for N3, N50 and OA on the ox-VOC yields within the 6-D parameter space. A weak dependency between an ox-VOC and model skill does not imply that the contribution of the ox-VOC to OA and particle number concentration is unimportant. Rather, it implies that within the current modelling framework its contribution can be compensated by changes in other ox-VOCs.*

*To identify the plausible and implausible parts of the parameter space using the patterns of dependencies, the ensemble simulations (denoted by triangles in Figures 10, 11 and 12) in the subplots are shaded blue to red. Darker shades of blue indicate low/poor Taylor skill score and darker shades of red represent high/good Taylor skill score within the ensemble. We note the relative rank of the simulations in Taylor skill score and their relative positions in each 2-D subplot and use these two information to identify clusters of blue or red triangles in the parameter space. For absolute values of Taylor skill scores of each simulation see Table A1 and A3, A4 and A5 (which are Figures 10, 11 and 12 labelled with simulation number).*

---

## Author Comment (AC2) · 18 Nov 2020

The modeling of secondary organic aerosol (SOA) formation involves enormous complexity. Sengupta et al. used a perturbed parameter ensemble of 60 model simulations to evaluate six representative SOA precursors on aerosol number concentration and organic aerosol mass. The model results have been evaluated against measurements from sites across the globe. The authors determined the critical role of ELVOC and LVOC in new particle formation and growth, and the importance of LVOC and SVOC in simulating organic mass. They also found a high degree of equifinality in the SOA model. Parameter combinations that are best for aerosol number concentration are worst for organic aerosol mass. This manuscript provides a novel approach to evaluate effects of parameter combinations, rather than individual parameters, on both aerosol number and mass concentration that matter for climate impacts. I would recommend publication of this manuscript after addressing minor issues below.

Thank you for the constructive feedback. Please see our replies inline. Any text added in the revised manuscript is shown below in *bold, italics*.

1. A brief introduction on the meaning and climate implications of N3 and N50 should be added somewhere in the introduction (not later in Page 20).

The following has been added as Section 2.3 in the revised manuscript.

**Microphysical** processes**

SOA formation in the model starts with B\_ELVOC and sulphuric acid via nucleation. Nucleation rates in the model (Kirkby2016, Gordon2016) determine the formation of clusters of 1.7 nm dry diameter. Thereafter their growth up to 3 nm sizes in the model is determined using the equation of (Kerminen2002) which takes into account the losses during initial growth of clusters. Clusters that reach a dry diameter of 3 nm are added to the nucleation mode as freshly nucleated particles. Thus N3, the number concentration of particles with dry diameter greater than 3 nm (in cm-3), represents the total particle number concentration in the model.

Once particles appear in the nucleation mode they may either grow using sulphuric acid and ox-VOCs (as described above) or get scavenged by larger particles acting as condensation sink. Particles that reach a dry diameter of 50 nm can act as cloud condensation nuclei in the atmosphere. Thus N50, the number concentration of particles with dry diameter greater than 50 nm (in cm-3), represents the number of climatic-relevant sized particles in the model.

Aerosol particles are removed through dry deposition, sedimentation, nucleation

scavenging and impact scavenging. Dry deposition accounts for gravitational settling, Brownian motion, impaction interception, particle rebound and predominantly removes particles smaller than 50 nm. Processes represented under wet deposition are nucleation scavenging and impact scavenging.

2. Some more information about the GLOMAP model is needed in Section 2.1, e.g. what are the emission inventories used here? How does the model treat the removal of aerosols? What do anthropogenic VOCs include?

Emission inventories are specified in Section 2.2, SOA scheme.

Model removal processes have now been added in Section 2.3 (see response to comment #1)

We use spatially co-located CO emissions to represent anthropogenic VOCs, specified in Section 2.2.

3. Table 2: B\_ELVOC does not produce any SOA in the default scheme? Why?

B\_ELVOC represents atmospheric extremely low volatile biogenic organic compounds and take part in the formation of nucleated clusters in the model. Once these clusters grow to 3nm (by condensation of sulphuric acid and B\_LVOC) they appear as SOA particles in the nucleation mode. The mass of the sub-3nm nucleated clusters produced by B\_ELVOC are negligible and hence, do not contribute to the SOA produced by the model.

4. Figure 2: Would you please say a bit more about how you design these 60 experiments? It may be clearer to include an appendix table to show the combination of yields you chose and explain why. At P14 L18 you said, "the total global production of SOA varies by only a factor of 4 and the lowest value is 220 Tg yr-1". This really depends on the choice of yields combination, right? If you choose 0 for all yields, the SOA would be zero? And from Figure 2, I did not see an experiment that uses maximum scaling/yields for all sources. Would such an experiment show a much higher SOA production than 850 Tg/yr?

We used the maximin Latin hypercube sampling method to design the 60 parameter combinations. This method works such that the uncertainty range for each parameter was divided into 60 bins and one point drawn from each bin for the combinations, so that no parameter value was repeated. The maximin Latin hypercube sampling method requires that each point drawn must be as far apart from the previous value as possible, ensuring maximum separation between ensemble members in the multivariate space. This statistical method ensured a good coverage of the 6-D parameter space by the ensemble.

SOA would span the range from zero to greater than 850 Tg/yr if we used the minimum

and maximum parameter combinations respectively for all the 6 ox-VOCs. But as discussed such parameter combinations do not satisfy the space-filling design criteria of the Latin hypercube sampling method. Further, such combinations (all ox-VOCs being 0 or maximum of the range) are next to impossible in the atmosphere and hence, not explored. Furthermore, evaluation against observations shows that 'reality' lies within the range of the Latin hypercube sample.

5. P13: Could you please provide more information on TSS? How did you calculate R and Ro? Is the normalized standard deviation calculated by all model vs observation monthly mean data across the globe? And could you give a more intuitive interpretation of the TSS values? Later in Figure 10-12 you defined "low/poor" versus "high/good" TSS, which should be explained here as well.

The following is added to the manuscript.

**R* is the Pearson correlation coefficient and *R*\_0 is the maximum correlation attainable by the model, assumed to be 1**

Yes, the normalized standard deviation is calculated with all model and observation data across the globe.

The following text is added in P14 L13-17 to help interpretation of TSS values.

As the model variance approaches the variance in the observations and \$R\$ approaches \$R\_0\$, *i.e. the model is most skilful*, \$TSS\$ approaches unity. As the model variance approaches zero or as the correlation coefficient between model and observation becomes more negative, TSS approaches zero. TSS thus takes into account both how well the model simulates the observed pattern (correlation coefficient) as well as how close model observation agreement is (variance). The full statistics (TSS, NMBF and R) calculated for each simulation within the ensemble is presented in Table. A1.

Good and poor TSS is explained in P27 L13.

To identify the plausible and implausible parts of the parameter space using the patterns of dependencies, the ensemble simulations (denoted by triangles in Figures 10, 11 and 12) in the subplots are shaded blue to red. Darker shades of blue indicate low/poor Taylor skill score and darker shades of red represent high/good Taylor skill score within the ensemble. We note the relative rank of the simulations in Taylor skill

score and their relative positions in each 2-D subplot and use these two information to identify clusters of blue or red triangles in the parameter space. For absolute values of Taylor skill scores of each simulation see Table. A1 and Figures A3, A4 and A5 (which are Figures 10, 11 and 12 labelled with simulation number).

6. P14 L4: "The global mass of SOA produced in the model simulations ranges from 220 to 850 Tg yr-1". Doesn't this include the 107 Tg/yr of SOA produced by the default SOA scheme (based on Table 2)?

That is correct, the SOA range of this ensemble doesn't include the values produced by the default SOA scheme. As discussed in comment #4, the maximin Latin hypercube method was used to create the space-filling design of experiments. All parts of the 6-D parameter space were explored equally with no additional weighting given to the model's default parameter space.

There could be numerous other possible combinations of ox-VOCs (apart from the 60 combinations explored here) and some of those combinations could produce N3, N50 and SOA outside the range of that produced by this ensemble. The parameter combination of ox-VOCs in the default SOA scheme is one such example.

This study (based on comparing the perturbed parameter ensemble with observations) concludes on the plausible and implausible ranges for the six perturbed parameters, not the model outputs i.e. we do not provide an estimate of modeled global mass of SOA to be between 220-850 Tg/yr. We simply point out that our SOA range overlaps with that found in other studies (Spracklen et al., 2011).

7. P19 L18: Why do high B\_SVOC\_I and A\_LVOC suppress particle growth?

Because of enhanced condensation of vapours on larger accumulation and coarse mode particles caused by the additional SVOC and LVOC.

The following text explaining this has been added to P20 L7.

"B\_SVOC\_I, A\\_LVOC, and A\\_SVOC, are not spatially co-located with the nucleated clusters produced from biogenic ox-VOCs and facilitate the growth of larger particles which then increases the coagulation sink for nucleated clusters and nucleating vapours, thereby effectively suppressing the growth of nucleated clusters to N50-relevant sizes in the above simulations with low B\_LVOC yields."

8. Caption of Figure 7 is wrong. N3 is not shown. Q2 and IQR not consistent.

Caption of Figure 7 is corrected.

9. Figure 7: Could the "global mean" OA be as high as 3-10 ug/m3? Did you do an areaaveraged OA over the whole globe (including ocean), or the total land area, or averaged over the available sites?

The global mean for each simulation is calculated over the land area. The distribution of OA for each simulation is shown in Figure 4 and the global mean values printed.

10. P20 L13-14 (the first sentence actually) should refer to Figure 7.

Reference to Figure 7 is added to P21 L12.

11. P20 L21-22 Did Figure 7 include any information of nucleation? "Including new, more accurate nucleation pathways into models is unlikely to improve the model performance with respect to N50." This sentence is confusing.

Reference to Figure 7 is removed from this sentence. And the entire paragraph is moved to P20 L20-27 in the revised manuscript before the discussion on Figure 7.

12. P20 The whole paragraph "OA concentrations in Figure 7 are found to be unrelated to B\_ELVOC..." It could be helpful to include regression lines and some statistics (e.g. slope, significance, correlations) to support your statements here.

We think the conclusions we draw from Figure 7 do not warrant precise statistics. We simply point out clearly visible trends (such as B\_ELVOC versus N50) in contrast with scatter (such as B\_ELVOC versus OA) between the ox-VOC parameters and each model output. Some ox-VOCs have more than one relationship trends with the model outputs varying across the full parameter range. We think too much additional statistics on the ensemble may distract from the main statistics i.e. the statistical summary of comparison of the ensemble with observations of N3, N50 and OA.

13. P21 "This is because A\_LVOCs or A\_SVOCs grow fewer particles than their biogenic counterparts and as a result changes in their concentrations have a lesser impact on simulated OA mass." Again, is such information included in Figure 7? Is it possible that it's just because anthropogenic sources produce much less SOA than biogenic sources on the global scale?

The following text has been added in P22 L5 to explain this point.

This is because *in the current SOA scheme* A\_LVOCs or A\_SVOCs grow fewer particles than their biogenic counterparts *which have the same spatial distribution as the nucleated particles they produce.*

Yes that is correct. If future studies discover and/or quantify a greater role of anthropogenic VOCs on SOA, including role in nucleation and cluster growth, this will change their impact on the model predicted OA concentrations.

14. P23 L24 "the model overestimates the strength of seasonal cycle". This is only true for just a few sites, right?

More explanation and plots have been added in the Appendix to show this holds true in general for the model, with exceptions in some sites (e.g. Hyytiala). We think this is because anthropogenic SOA sources (which peak in the winter) and their role in particle formation/growth are under-represented in the model.

15. Why not include color bars to show the TSS range in Figure 10-12?

The following text explaining the above is added P27 L14 in the revised manuscript.

To identify the plausible and implausible parts of the parameter space using the patterns of dependencies, the ensemble simulations (denoted by triangles in Figures 10, 11 and 12) in the subplots are shaded blue to red. Darker shades of blue indicate low/poor Taylor skill score and darker shades of red represent high/good Taylor skill score within the ensemble. We note the relative rank of the simulations in Taylor skill score and their relative positions in each 2-D subplot and use these two information to identify clusters of blue or red triangles in the parameter space. For absolute values of Taylor skill scores of each simulation see Table A1 and A3, A4 and A5 (which are Figures 10, 11 and 12 labelled with simulation number).

16. P26 L17-26. These interpretations are very confusing, and Figure 10-12 are hard to understand. For example:

The following text on how to interpret Figures 10-12 is added P26 L6 in the revised manuscript.

Each scatter plot in Figures 10, 11 and 12 show the relationship between two ox-VOCs for each of the sixty ensemble members. Note the values for B\_ELVOC yields shown in Figures 10, 11 and 12 and Figures A3, A4 and A5 are scaling factors which have to be multiplied to the baseline model yields (Table 2) to get the B\\_ELVOC yields for the ensemble members. For the rest of the ox-VOCs the values shown are yield values in % which can be converted to Tg yr-1 using Table 2. Because it is a 6-D space, it is also important to note that the other four parameters are varying randomly across each plane.

We use Figures 10, 11 and 12 to identify patterns of dependencies of the Taylor skill

scores for N3, N50 and OA on the ox-VOC yields within the 6-D parameter space. A weak dependency between an ox-VOC and model skill does not imply that the contribution of the ox-VOC to OA and particle number concentration is unimportant. Rather, it implies that within the current modelling framework its contribution can be compensated by changes in other ox-VOCs.

Also see response to comment #15 for more information on how to interpret the Figures 10-12.

"where B\_ELVOC yield is greater than 19.8%": Does this refer to first column in Figure 10?

Yes, that's right. The following text has been added to help the reader.

We note two additional regions in the 6-D space dominated by low model skill in N3 - where B\_ELVOC yield is greater than 19.8\% and B\_LVOC is less than 113 Tg yr (*Figure 10 first column, second row. Bottom right corner of the subplot corresponding to 6 times the baseline yield in the x-axis and a yield of 100\% in the y-axis) and where the sum of anthropogenic LVOC and SVOC is greater than 127 Tg yr\$^{-1}\$ (Figure 10 fifth column, sixth row. Top right corner of the subplot corresponding to 200\% yield in both x- and y-axes).*

More references to specific subplots and figures are also added throughout this section for easier interpretation.

"B\_LVOC is less than 113 Tg yr-1 (corresponding to a yield of 100% in Figure 10 first column, second row from top)": I think this should refer to second column in Figure 10? The "first column, second row from top" refers to the scatter plot with B\_ELVOC on the x-axis and B\_SVOC\_M on the y-axis. I don't think this is the one you are referring to here.

Stands correct with the change in Figures.

"where the sum of anthropogenic LVOC and SVOC is greater than 127 Tg yr-1 (200% yield in Figure 10 fifth column, sixth row)." – There is no sixth row in Figure 10. Should be fifth row?

Stands correct with the change in Figures.

17. P26 Second paragraph about N50. Again, "Figure 11 first column, second row" refers to the scatter plot with B\_ELVOC on the x-axis and B\_SVOC\_M on the y-axis, not consistent with what you are saying here.

Stands correct with the change in Figures.

18. Figure 10-12 may need to be rearranged for an easier interpretation. For example,

both X and Y axis should include 6 ox-VOCs and in the same order, i.e. have B\_ELVOC, B\_LVOC, ... A\_SVOC on the X-axis from left to right, and on the Y-axis from bottom to top. Also please define "good" and "bad" TSS scores.

We have modified Figures 10-12 for an easier interpretation. There are now 6 panels vertically and horizontally in the scatterplot matrix. Text on how to read the figures have been added – see response to comments #15 and #16.

All Taylor Skill Scores are listed in Table A1. Figures 10-12, are used to identify patterns of dependencies of the scores on specific regions of the parameter spaces. We use the shading in these plots to identify clusters of blue and red triangles, the exact absolute values of scores for each simulation in Figs 10-12 can be found from Table A1 and Figures A3-A5. Text explaining this has been added to the revised manuscript – see response to comment #15..

19. P31 Second line "... for all five ox-VOCs in 1-D": should be "six ox-VOCs" here.

Corrected.

20. P34 I like the statements about B\_LVOC, B\_SVOC\_I/M, A\_SVOC which are much clearer than in the results section, but would you please refer back to the figures from which you draw those conclusions? In general Section 3.2 is hard to follow with quite a few typos/errors and Figure 10-12 are not easy to interpret. Please check through the manuscript for potential mismatches and typos.

References to figures are added in the conclusion. Such as in P35 L22 and P36 L9.

B\_ELVOC strongly influences model skill scores in N3 and to a lesser extent in N50 *(Figure 13).*

B\_LVOC has the strongest influence on model skills in N3, N50 and OA (second column, Figures 10, 11 and 12)